METHODS

# CTFacTomo: Reconstructing 3D spatial structures of RNA tomography transcriptomes by collapsed tensor factorization

Tianci Song, Quoc Nguyen, Charles Broadbent, Rui Kuang *

Department of Computer Science and Engineering, University of Minnesota Twin Cities, Minneapolis, Minnesota, United States of America

* kuang@umn.edu

## Abstract

Cells are organized to form three-dimensional structures of complex tissues. To map the complete 3D organization of a tissue, technologies based on tissue microdissections provide deep bulk RNA sequencing of orthogonally arranged cryosections of a tissue, such that the full 3D spatial structure could be inferred from deeply sequenced transcriptomes in three views projected similarly as 3D tomography. Here, we introduce CTFacTomo to learn a Collapsed Tensor Factorization for RNA tomography data from cryosections to reconstruct 3D spatially resolved gene expressions. CTFacTomo combines tensor factorization with collapsing tensor entries to match the bulk gene expressions in each cryosection, enriched by a regularization of a product graph of protein-protein interaction network and spatial graphs. In the experiments, CTFacTomo is first validated on three datasets projected from fully profiled 3D spatial gene expressions to demonstrate that CTFacTomo significantly outperforms the benchmark methods for predicting the ground-truth gene expressions based on the projected 1D spatial gene expressions of three orthographic views. CTFacTomo is then applied to two RNA tomography datasets from zebrafish embryo and mouse olfactory mucosa, respectively. In both datasets, CTFacTomo detects 3D spatial expressions of several marker genes that are consistent with the developmental or functional regions in comparison to accompanying ISH staining images. In addition, a qualitative comparison between the reconstructed zebrafish embryo gene expressions with a matched external 3D Stereo-seq dataset also suggests that CTFacTomo reconstructs more spatially coherent patterns in the whole transcriptome with state-of-the-art performance.

### Author summary

In multicellular organisms, the three-dimensional organization of cells is fundamental to forming complex tissue architectures to support essential cellular

**Data availability statement:** CTFacTomo is implemented in Python and the code is publicly available through GitHub at https://github.com/kuanglab/CTFacTomo. The datasets analyzed in this study are available in raw form from their original sources. Specifically, the 3D spatial transcriptomics data used to simulate RNA tomography are derived from either low-resolution ST1K or high-resolution Stereo-seq platforms, where data for the human heart tissue were obtained from the project website https://data.mendeley.com/datasets/mbvhhf8m62/2, mouse brain data were collected from https://molecularatlas.org/download-data, and Drosophila embryo data are accessible at https://db.cngb.org/stomics/flysta3d/. Tomo-seq data for 3D spatial transcriptome reconstruction from the zebrafish embryo at the shield stage, along with the ISH images of marker genes, are available at https://www.ncbi.nlm.nih.gov/geo/query.cgi?acc=GSE59873, while the Stereo-seq data of the zebrafish embryo at the shield stage are available at https://db.cngb.org/stomics/zesta/download/. The Tomo-seq data from mouse olfactory mucosa and ISH images of marker genes can be accessed at https://www.ebi.ac.uk/biostudies/arrayexpress/studies/E-MTAB-10211.

**Funding:** This research work is supported by a grant from the National Science Foundation, USA (NSF BIO DBI-IIBR 2042159 to RK). The funders had no role in study design, data collection and analysis, decision to publish, or preparation of the manuscript.

**Competing interests:** The authors have declared that no competing interests exist.

functions such as cell movement, communication and interactions, and spatially varying gene expression. RNA tomography technology applies 1D transcriptomic profiling to consecutive slices along orthogonal spatial axes of tomography, and is particularly useful for capturing such 3D organization. CTFacTomo is a computational method based on collapsed tensor factorization for reconstructing 3D spatial gene expression from RNA tomography data. By learning a tensor decomposition from collapsed entries along three orthogonal views, CTFacTomo accurately identifies underlying 3D spatial transcriptomic patterns using the factorization model. CTFacTomo is first validated on three datasets projected from fully profiled 3D spatial gene expression data, demonstrating accurate recovery of ground-truth expressions from 1D projections of three orthogonal views. It is then applied to reconstruct 3D spatial gene expression in large-scale zebrafish embryo and mouse olfactory mucosa datasets, with results evaluated using ISH images and an external zebrafish Stereo-seq dataset. Together, these results demonstrate the superior performance of CTFacTomo for reconstructing 3D spatial gene expression from RNA tomography data.

## Introduction

Cells in multicellular organisms are organized to form complex 3D tissue structures and create distinct microenvironments to drive essential events such as cell movement, cell communication and interactions, and spatially varying gene expressions. For example, morphogen gradients diffuse through 3D space to regulate cell differentiation and morphogenesis, orchestrating the development of functional organs during embryogenesis. To reveal the spatial organization of the cells and their functions in the tissues, spatial transcriptomic technologies such as in-situ hybridization (ISH) [1–7] or in-situ capturing (ISC) [8–11] have been widely used to profile gene expression with retained spatial localization information in the tissue [12]. Among these spatial profiling technologies, technologies based on microdissection adopt a different strategy, which isolates slices or regions of tissues for separate RNA sequencing such that the location of the cells is informed by the organization of the arranged slices or regions [13]. Transcriptome tomography sequencing (Tomo-seq) [14] is a notable method in this category inspired by image tomography. In Tomo-seq, replicates of a 3D tissue are dissected into successive slices along three orthogonal views (axes), and then the RNAs in the cells in each slice are collected for deep bulk RNA sequencing. Finally, the 3D structures of the gene expressions could be reconstructed by the gene expressions measured from the slices stacked in each of the three axes using a computational method.

Microdissection-based methods provide more reliable transcriptome-wide gene expression measurements through highly sensitive bulk RNA sequencing, effectively avoiding key limitations, namely that ISC-based methods generally suffer from low RNA capture rates, while ISH-based methods are limited by lower detection sensitivity in transcriptome-wide profiling. More importantly, since both ISH and ISC

inherently profile RNAs in a 2D space, a challenging step is required to align stacked slices by a manual-intensive registration of annotated regions, tissue histological images, or spatial coordinates for 3D construction. In contrast, microdissection-based methods can perform RNA sequencing of slices dissected in different views from replicates for convenient 3D measuring of the gene expressions for full 3D reconstruction. In addition, the cost for library preparation and sequencing of cryosection samples are significantly lower than using fluorescent-imaging or spatial-array based profiling of 3D tissues in typical experiments. Despite these advantages, one significant limitation of Tomo-seq and other microdissection-based methods is that the reconstruction of the 3D spatial gene expression is a computational challenge and setback to a great extent. The current solution for 3D reconstruction with Tomo-seq is the iterative proportional fitting (IPF) algorithm [15], which is a relatively simplistic heuristic optimization method for minimizing fitting error for each individual gene independently. Accordingly, IPF is prone to noise in RNA sequencing data and might fail to capture complex spatial patterns [16].

In this paper, we introduce Collapsed Tensor Factorization for Tomography (CTFacTomo) to learn a tensor factorization representation of spatially resolved 3D gene expressions from RNA tomography data of cryosections. CTFacTomo reconstructs 3D spatial transcriptomics based on the 1D bulk RNA sequencing data obtained for the consecutive slices in each orthogonal spatial axis of tomography along the tissue as outlined in Fig 1. The 1D gene expression data represent the expression of each gene in each consecutive slice in one orthogonal axis (Fig 1A). Then, the 1D data in three views is combined to reconstruct the 3D spatial transcriptomes in a 4-way tensor with x-y-z spatial modes and gene mode as shown in Fig 1B. CTFacTomo combines tensor factorization with a loss function supervised by collapsing the tensor entries to match the gene expressions in each cryosection and the 3D tissue mask. CTFacTomo outputs a Canonical Polyadic decomposition of the reconstructed tensor aligned with the spatial axes. CTFacTomo also incorporates spatial relations among the slices along different axes and gene functional relations among genes with regularization by a product of a protein-protein interaction (PPI) network and spatial graphs to utilize important prior information (Fig 1B). 3D spatial expressions can be reconstructed from the decomposition model for identifications of 3D spatial patterns of marker genes and other spatial characteristics of the transcriptome (Fig 1C).

In the experiments, CTFacTomo is first validated on three datasets projected from fully profiled 3D spatial gene expressions to demonstrate that CTFacTomo can accurately predict the ground-truth gene expressions based on the projected 1D spatial gene expressions of three orthographic views. Then, CTFacTomo was applied to reconstruct 3D spatially resolved gene expressions in two large-scale 3D RNA tomography datasets, one from a zebrafish embryo and the other from a mouse olfactory mucosa, as shown in Fig 1C. The reconstructed spatial expressions are evaluated by comparisons with the accompanying ISH staining images of marker genes. Finally, an external Stereo-seq dataset of zebrafish embryo is also matched and used for the evaluation of the reconstructed gene expressions from the RNA tomography data.

## Materials and methods

In CTFacTomo, we model the reconstructed 3D spatial gene expression as 4-way tensor $\mathcal{T} \in \mathbb{R}_+^{n_g \times n_x \times n_y \times n_z}$, and we also model the 1D spatial gene expressions along different spatial axes as 2D matrices $\mathbf{X}_x \in \mathbf{R}_+^{n_g \times n_x}, \mathbf{X}_y \in \mathbf{R}_+^{n_g \times n_y}, \mathbf{X}_z \in \mathbf{R}_+^{n_g \times n_z}$, and the tensor-based model for 3D spatial gene expression reconstruction is illustrated in the Fig 1. The key modeling ideas are: A) the collapse of the reconstructed 3D spatial gene expression tensor $T$ along the gene and a given spatial axes collapse($\mathcal{T}, g, i$) should be identical to the 1D gene expression matrix $\mathbf{X}_i$; B) the reconstructed tensor $\mathcal{T}$ should be approximated by $\hat{\mathcal{T}}$ that can be compressed in a rank-$R$ CP decomposition form $\hat{\mathcal{T}} = [\![\mathbf{A}_g, \mathbf{A}_x, \mathbf{A}_y, \mathbf{A}_z]\!]$ for time and space efficiencies; C) the entries in the reconstructed tensor $\hat{\mathcal{T}}$ should align the spatial relations in the 3D space and functional relations among genes.

## Preliminaries

Here, we only present the minimal necessary definitions for introducing CTFacTomo.

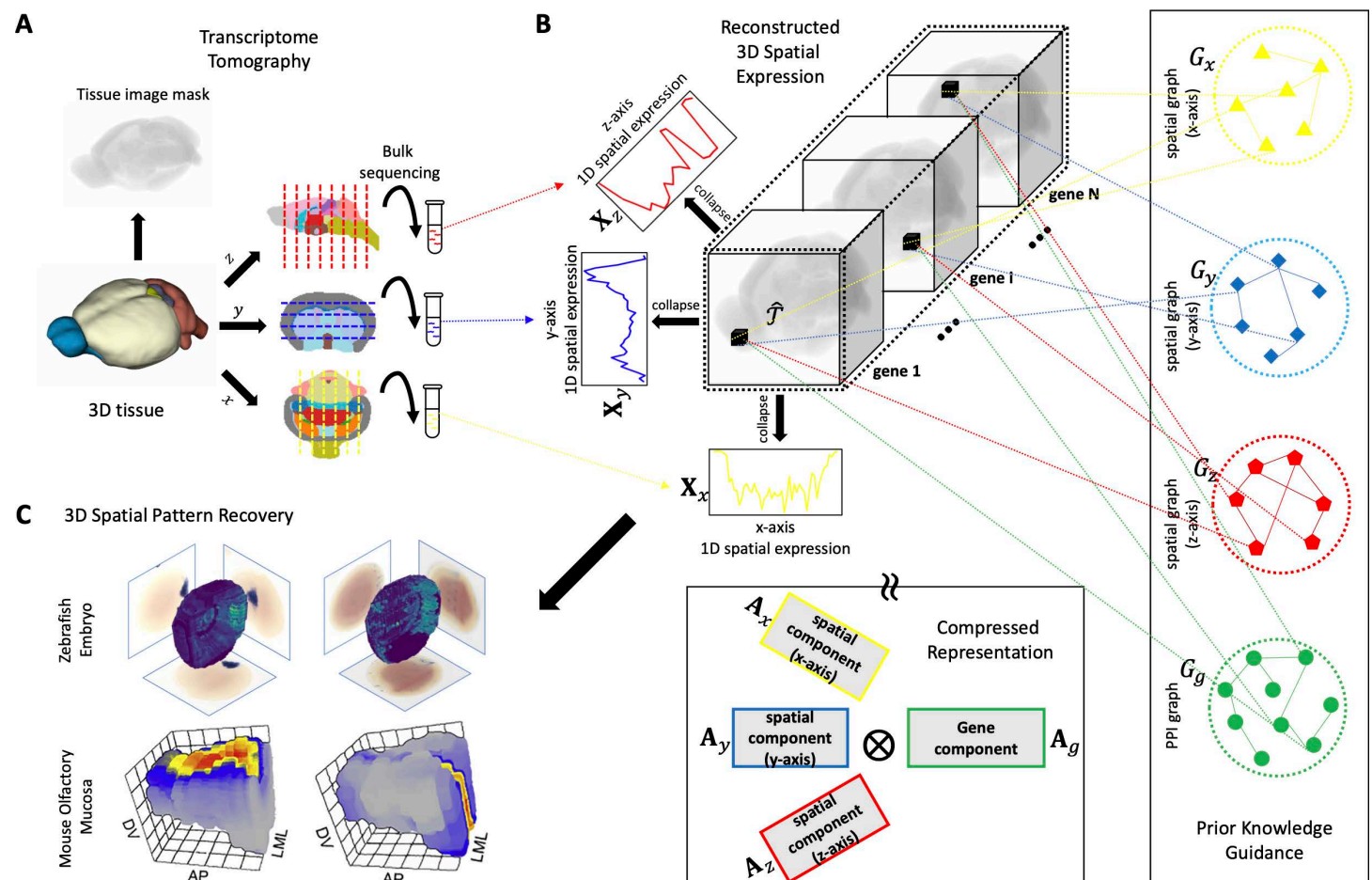

**Fig 1. Overview of CTFacTomo. A**: Transcriptome tomography generates high-coverage 1D spatial transcriptomics along three orthogonal spatial axes. A 3D tissue and its replicates are sectioned into successive slices along each axis, and bulk transcriptomics techniques are applied to quantify gene expression along the axis. Typically, a 3D mask of the profiled tissue can also be constructed from imaging. **B**: Given 1D spatial gene expression of each gene along three orthogonal spatial axes ($\mathbf{X}_x, \mathbf{X}_y, \mathbf{X}_z$), CTFacTomo reconstructs 3D spatial gene expression as a 4-way tensor $\hat{\mathcal{T}}$. The reconstruction is estimated via CP decomposition with four factor matrices ($\mathbf{A}_x, \mathbf{A}_y, \mathbf{A}_z, \mathbf{A}_g$), minimizing the reconstruction error with respect to the observed projections and the 3D tissue mask. CTFacTomo also incorporates prior knowledge from spatial graphs ($G_x, G_y, G_z$) and a protein-protein interaction (PPI) network $G_g$ to guide the 3D expression reconstruction. **C**: The reconstructed 3D spatial expression enables identifications of 3D spatial patterns of marker genes and other spatial characteristics of the transcriptome. Examples are shown for two large-scale 3D Tomo-seq datasets, one of zebrafish embryo and the other of mouse olfactory mucosa.

**Canonical Polyadic Decomposition (CPD).** An $M$-way tensor $\mathcal{T}$ can be approximated in a compressed rank-$R$ CPD representation as follows,

$$\mathcal{T} \approx \llbracket \mathbf{A}_1, \mathbf{A}_2, \ldots, \mathbf{A}_M \rrbracket = \sum_{r=1}^{n_r} \mathbf{a}_1^r \odot \mathbf{a}_2^r \odot \ldots \odot \mathbf{a}_M^r, \tag{1}$$

where $\llbracket \cdot \rrbracket$ denotes the Kruskal operator, $\odot$ denotes vector outer product, $n_r$ is the rank of the decomposition, $\mathbf{A}_i \in \mathbb{R}^{n_i \times n_r}$ represents the factor matrix along mode-$i$, and $\mathbf{a}_i^r$ is the vector in the $r$-th column of $\mathbf{A}_m$.

**Tensor matricization.** A tensor represented by CP decomposition can be matricized along $i$-th mode as $\mathcal{T}_{(i)} = \mathbf{A}_i(\mathbf{A}_1 \odot \ldots \odot \mathbf{A}_{i-1} \odot \mathbf{A}_{i+1} \odot \ldots \odot \mathbf{A}_M)^T$, where $\odot$ denotes Khatri-Rao product, and $\mathcal{T}_{(i)} \in \mathbb{R}^{n_i \times \prod_{j \neq i} n_j}$. As a special case of matricization, a tensor in the form of CP decomposition can be vectorized as $\text{vec}(\mathcal{T}) = (\mathbf{A}_1 \odot \ldots \odot \mathbf{A}_j \odot \ldots \odot \mathbf{A}_M)\mathbf{1}^T$, where $\text{vec}(\cdot)$ represents the vectorization function and $\mathbf{1}$ is an all-ones vector of size $n_r$.

**Tensor collapsing.** Collapsing operation collapses a tensor into a lower-order one with given modes by summing over entries along the remaining modes. Collapsing any one mode on the CPD of a tensor can be written as a collapsing function $\text{collapse}(\cdot)$ as follows,

$$\text{collapse}(\llbracket \mathbf{A}_1, \ldots, \mathbf{A}_M \rrbracket, i) =$$
$$\llbracket \mathbf{a}_1, \ldots, \mathbf{a}_{i-1}, \mathbf{A}_i, \mathbf{a}_{i+1}, \ldots, \mathbf{a}_M \rrbracket, \tag{2}$$

where $\mathbf{a}_i^T = \mathbf{1}_i^T \mathbf{A}_i$ denotes the row summation of $\mathbf{A}_i$. Similarly, collapsing given any two modes on the CPD of a tensor can be written as,

$$\text{collapse}(\llbracket \mathbf{A}_1, \ldots, \mathbf{A}_M \rrbracket, i, j) =$$
$$\llbracket \mathbf{a}_1, \ldots, \mathbf{a}_{i-1}, \mathbf{A}_i, \mathbf{a}_{i+1}, \ldots, \mathbf{a}_{j-1}, \mathbf{A}_j, \mathbf{a}_{j+1}, \ldots, \mathbf{a}_M \rrbracket. \tag{3}$$

Accordingly, the collapsing operation can be defined on the tensor of CP decomposition form given more than two modes.

**Product graph.** Given $M$ undirected graphs $\{G_m = (V_m, E_m) : m = 1, 2, \ldots, M\}$, where $V_m$ and $E_m$ denote the set of nodes and edges in the graph $G_m$, and $|V_m| = n_m$ denotes the number of nodes in the $G_m$. Let $G_p$ be a new graph combining the set of graphs $\{G_m : m = 1, 2, \ldots, M\}$, denoted as product graph $G_p = (V_p, E_p)$ with the number of nodes $|V_p| = \prod_{m=1}^M |V_m|$. For any pair of nodes $(a_1, a_2, \ldots, a_M)$ and $(b_1, b_2, \ldots, b_M)$ in the product graph $G_p$, the corresponding edge is determined by the adjacency or equality of $a_m$ and $b_m$ in the graph $G_m$. For simplicity, we only used Cartesian product graph in this work, where $((a_1, a_2, \ldots, a_M), (b_1, b_2, \ldots, b_M)) \in E_p$ if and only if $(a_m, b_m) \in E_m$ while $a_k = b_k, \forall k \neq m$.

Let $\mathbf{W}_m$ be the adjacency matrix of $G_m$, where $[\mathbf{W}_m]_{ij} = 1$ if there is an edge between $i$- and $j$-th nodes in $G_m$ and $0$ otherwise, and let $\mathbf{D}_m = \text{diag}(d_1, \ldots, d_{n_m})$ be the degree matrix of $G_m$ with $d_{ij} = \sum_j [\mathbf{W}_m]_{ij}$. $\mathbf{L}_m = \mathbf{D}_m - \mathbf{W}_m$ represents the graph Laplacian for $G_m$. Then the adjacency matrix and Laplacian matrix of Cartesian product graph $G_p$ can be calculated by $\mathbf{W}_p = \oplus_{m=1}^M \mathbf{W}_m$ and $\mathbf{L}_p = \oplus_{m=1}^M \mathbf{L}_m$ respectively, where $\oplus$ denotes the Kronecker sum.

**Product graph Laplacian regularization.** Suppose a $M$-way tensor $\mathcal{T}$ with each mode $m$ associated with a graphs $G_m$ encoding the prior knowledge. Let $G_p$ be the product of the graphs from different modes, where $\mathbf{W}_p$ and $\mathbf{L}_p$ are adjacency matrix and Laplacian matrix of $G_p$. Each entry in $\mathcal{T}$ corresponds to one node in $G_p$, and then graph Laplacian regularization can be used to smooth values in $\mathcal{T}$ over the manifolds of the graph $G_p$, ensuring that adjacent nodes in this high-order graph share similar values, which can be mathematically defined in the quadratic form as $\text{vec}(\mathcal{T})^T \mathbf{L}_p \text{vec}(\mathcal{T}) = \sum_{ij} [\mathbf{W}_p]_{ij} ([\text{vec}(\mathcal{T})]_i - [\text{vec}(\mathcal{T})]_j)$. When $\mathcal{T}$ is represented using the form of CP decomposition, product graph Laplacian regularization can be formulated as follows:

$$\text{vec}(\mathcal{T})^T \mathbf{L}_p \text{vec}(\mathcal{T}) = \text{vec}(\mathcal{T})^T (\oplus_{m=1}^M \mathbf{L}_p) \text{vec}(\mathcal{T})$$
$$= \sum_{m=1}^M \mathbf{1}^T \left( (\mathbf{A}_m^T \mathbf{L}_m \mathbf{A}_m) \circledast_{k \neq m} (\mathbf{A}_k^T \mathbf{A}_k) \right) \mathbf{1}, \tag{4}$$

where $\circledast$ denotes Hadamard product, and $\mathbf{1}$ is an all-ones vector of size rank $n_r$.

## Loss function

The overview of CTFacTomo is illustrated in Fig 1. The inputs of CTFacTomo are the tomography transcriptomics data along different spatial axes as gene expression matrices $\mathbf{X}_x \in \mathbf{R}_+^{n_g \times n_x}, \mathbf{X}_y \in \mathbf{R}_+^{n_g \times n_y}, \mathbf{X}_z \in \mathbf{R}_+^{n_g \times n_z}$, and a 3D image mask as a *3*-way tensor $\mathcal{M} \in \{0,1\}^{n_x \times n_y \times n_z}$ as shown in Fig 1A. The 1D spatial chain relations along three orthogonal axes and functional relations among genes are given in four knowledge graphs $\{G_i, i = g, x, y, z\}$ (Fig 1C). CTFacTomo reconstructs 3D spatial gene expression in a *4*-way tensor $\hat{\mathcal{T}} \in \mathbb{R}_+^{n_g \times n_x \times n_y \times n_z}$ represented by CP decomposition while being guided by prior knowledge encoded in the spatial and functional graphs (Fig 1B). We next formulate the learning problem and define each term in the loss function of CTFacTomo.

*Reconstruction loss.* We first define reconstruction loss $\mathcal{J}_1$, which ensures the collapse of the reconstructed 3D spatial gene expression tensor $\hat{\mathcal{T}}$ along the gene and a given spatial axes should be identical to the 1D gene expression matrix $\mathbf{X}_i$, and the loss term can be written as,

$$\mathcal{J}_1 = \frac{1}{2} \sum_{i=\{x,y,z\}} \left\| \mathbf{X}_i - \mathrm{collapse}(\hat{\mathcal{T}}, g, i) \right\|_F^2.$$

(5)

Here, 4-way tensor $\hat{\mathcal{T}}$ is collapsed (summed) over the other two spatial modes along each of the x-axis, y-axis and z-axis, exactly modeling the tomography projection onto the cryosections in each view.

*3D Masking loss.* To prevent the reconstructed 3D spatial gene expression from overspreading to regions outside of the tissue, a loss term $\mathcal{J}_2$ is introduced to fit the given 3D mask $\mathcal{M}$ that outlines the tissue shape as follows,

$$\mathcal{J}_2 = \frac{1}{2} \left\| \mathcal{M} \circledast \mathrm{collapse}(\hat{\mathcal{T}}, x, y, z) \right\|_F^2,$$

(6)

where $\circledast$ denotes Hadamard product. Here, the 4-way tensor $\hat{\mathcal{T}}$ is collapsed (summed) over all the genes in each spatial location (spot), and the loss penalizes the non-zero sums in the spots outside of the tissue.

*Product graph Laplacian regularization.* To leverage both spatial relations in 3D space and functional relations among genes for reconstruction, a Laplacian regularization of the Cartesian product of gene and spatial graphs $\{G_i, i = g, x, y, z\}$ in $\mathcal{J}_3$ is used to smooth the entries in $\mathcal{T}$ over the manifolds in the Cartesian product graph $G_p$, ensuring that the expressions of adjacent nodes in this high-order graph share similar values, defined as follows,

$$\mathcal{J}_3 = \frac{1}{2} \mathrm{vec}(\hat{\mathcal{T}})^T \mathbf{L}_p \mathrm{vec}(\hat{\mathcal{T}}).$$

(7)

Here $\mathbf{L}_p = \mathbf{L}_g \oplus \mathbf{L}_x \oplus \mathbf{L}_y \oplus \mathbf{L}_z$ is the graph Laplacian of $G_p$, where $\mathbf{L}_i, \forall i = g, x, y, z$, is the graph Laplacian of $G_i$, and $\oplus$ denotes Kronecker sum [17]. Note that there are important computational considerations in using chain graphs along the three spatial axes. The Cartesian product of these three chain graphs forms a 3D grid graph among the spots. This simplified representation addresses the scalability challenges of applying graph Laplacian regularization to large 3D spatial graphs containing tens of thousands of spots. This approach has proven effective for 2D spatial transcriptomics reconstruction in our previous work [18,19], and here we extend the formulation naturally to 3D using the same general framework. Moreover, the graph Laplacian regularization of the product graph, which incorporates both the chain graphs and the PPI network, captures global relationships among spatial locations beyond immediate neighbors. As a result, explicit modeling of more distant neighbors is generally unnecessary, except when more fine-grained customization is required.

*Total loss function.* Lastly, the total loss function $\mathcal{J}$ to learn a rank-$R$ CP decomposition $\hat{\mathcal{T}} = [\![\mathbf{A}_g, \mathbf{A}_x, \mathbf{A}_y, \mathbf{A}_z]\!]$ is the sum of $\mathcal{J}_1, \mathcal{J}_2$ and $\mathcal{J}_3$ as follows,

$$\min_{\{\mathbf{A}_g, \mathbf{A}_x, \mathbf{A}_y, \mathbf{A}_z\}} \mathcal{J} = \mathcal{J}_1 + \alpha\mathcal{J}_2 + \beta\mathcal{J}_3$$

$$\mathcal{J}_1 = \frac{1}{2}\sum_{i=\{x,y,z\}} \left\|\mathbf{X}_i - \text{collapse}([[\mathbf{A}_g, \mathbf{A}_x, \mathbf{A}_y, \mathbf{A}_z]], g, i)\right\|_F^2$$

$$\mathcal{J}_2 = \frac{1}{2}\left\|\mathcal{M} \circledast \text{collapse}([[\mathbf{A}_g, \mathbf{A}_x, \mathbf{A}_y, \mathbf{A}_z]], x, y, z)\right\|_F^2$$

$$\mathcal{J}_3 = \frac{1}{2}\text{vec}([[\mathbf{A}_g, \mathbf{A}_x, \mathbf{A}_y, \mathbf{A}_z]])^T \mathbf{L}_p \text{vec}([[\mathbf{A}_g, \mathbf{A}_x, \mathbf{A}_y, \mathbf{A}_z]]),$$

(8)

where $\alpha$ and $\beta$ are hyperparameters weighting the loss terms in the loss function.

## Multiplicative updating

To minimize the loss function in Eq (8), we propose an iterative optimization method based on multiplicative updating rules [20–22], which ensures a stationary solution of $\hat{\mathcal{T}}$ by alternatively updating each $\{\mathbf{A}_i, i = g, x, y, z\}$ with the derivatives of $\mathcal{J}$ with respect to the $\mathbf{A}_i$.

For simplicity, we decompose the loss function into multiple terms and introduce their derivatives with respect to $\mathbf{A}_g$ and $\mathbf{A}_i, \forall i = x, y, z$ individually. To facilitate the calculation of the derivatives, we define the auxiliary variables in Table 1.

With the definitions in Table 1, the partial derivatives of $\mathcal{J}_1$ with respect to $\mathbf{A}_g$ and $\mathbf{A}_i, \forall i = x, y, z$ can be represented as follows:

$$\frac{\partial\mathcal{J}_1}{\partial\mathbf{A}_g} = \mathbf{A}_g \sum_{i=\{x,y,z\}} \Phi_i^{(-g)} - \sum_{i=\{x,y,z\}} \Theta_i^{(-g)}$$

$$\frac{\partial\mathcal{J}_1}{\partial\mathbf{A}_i} = \mathbf{A}_i\Phi_g^{(-i)} + \mathbf{1}_i\mathbf{1}_i^T\mathbf{A}_i(\sum_{j\neq i}\Phi_{j,g}^{(-i)}) - \Theta_g^{(-i)} - \sum_{j\neq i}\Theta_{j,g}^{(-i)}.$$

(9)

Similarly, the partial derivatives of $\mathcal{J}_2$ with respect to $\mathbf{A}_g$ and $\mathbf{A}_i, \forall i = x, y, z$ can be represented as follows:

$$\frac{\partial\mathcal{J}_2}{\partial\mathbf{A}_g} = \mathbf{1}_g(\text{vec}(\mathcal{M}) \circledast \text{vec}([[\mathbf{a}_g, \mathbf{A}_x, \mathbf{A}_y, \mathbf{A}_z]]))^T \Upsilon^{(-g)}$$

$$\frac{\partial\mathcal{J}_2}{\partial\mathbf{A}_i} = (\mathbf{M}_{(i)} \circledast [[\mathbf{a}_g, \mathbf{A}_x, \mathbf{A}_y, \mathbf{A}_z]])\Upsilon_g^{(-i)},$$

(10)

where $\mathbf{M}_{(i)} \in \{0, 1\}^{n_i \times (\prod_{j\neq i} n_j)}$ denotes the mode-$i$ matricization of tensor $\mathcal{M}$.

**Table 1. Auxiliary variables.**

| Variable | Definition |
|---|---|
| $\Phi_j^{(-i)}$ | $\circledast_{k\neq i,j}(\mathbf{a}_k\mathbf{a}_k^T) \circledast (\mathbf{A}_j^T\mathbf{A}_j)$ |
| $\Phi_{j,k}^{(-i)}$ | $\circledast_{l\neq i,j,k}(\mathbf{a}_k\mathbf{a}_k^T) \circledast (\mathbf{A}_j^T\mathbf{A}_j) \circledast (\mathbf{A}_k^T\mathbf{A}_k)$ |
| $\Theta_j^{(-i)}$ | $\mathbf{X}_{(i,j)}(\odot_{k\neq i,j}\mathbf{a}_k^T \odot \mathbf{A}_j)$ |
| $\Theta_{j,k}^{(-i)}$ | $\mathbf{1}_i\text{vec}(\mathbf{X}_{(j,k)})^T(\odot_{l\neq i,j,k}\mathbf{a}_l^T \odot \mathbf{A}_j \odot \mathbf{A}_j)$ |
| $\Upsilon^{(-i)}$ | $\odot_{j\neq i}\mathbf{A}_j$ |
| $\Upsilon_j^{(-i)}$ | $\odot_{k\neq i,j}\mathbf{A}_k \odot \mathbf{a}_j^T$ |
| $\Psi_j^{(-i)}$ | $(\mathbf{A}_j^T\mathbf{L}_j\mathbf{A}_j) \circledast_{k\neq i,j} (\mathbf{A}_k^T\mathbf{A}_k)$ |

Following the derivation in [22], the partial derivatives of $\mathcal{J}_3$ with respect to $\mathbf{A}_i, \forall i = g, x, y, z$ can be represented as follows:

$$\frac{\partial \mathcal{J}_3}{\partial \mathbf{A}_i} = (\mathbf{L}_i \mathbf{A}_i)(\circledast_{j \neq i} \mathbf{A}_j^T \mathbf{A}_j) + \mathbf{A}_i(\sum_{j \neq i} \Psi_j^{(-i)})$$

$$= (\mathbf{D}_i \mathbf{A}_i)(\circledast_{j \neq i} \mathbf{A}_j^T \mathbf{A}_j) + \mathbf{A}_i(\sum_{j \neq i} \Psi_j^{(-i)}) -$$

$$(\mathbf{W}_i \mathbf{A}_i)(\circledast_{j \neq i} \mathbf{A}_j^T \mathbf{A}_j), \tag{11}$$

where $\mathbf{W}_i$ and $\mathbf{D}_i$ are the adjacency matrix and degree matrix of the graph $G_i$, and $\mathbf{L}_i = \mathbf{D}_i - \mathbf{W}_i$.

After combining the derivatives in Eqs (9), (10), (11), the derivative of $\mathcal{J}$ with respect to $\mathbf{A}_i, \forall i = g, x, y, z$ can be formalized as follows:

$$\frac{\partial \mathcal{J}}{\partial \mathbf{A}_i} = \frac{\partial \mathcal{J}_1}{\partial \mathbf{A}_i} + \frac{\partial \mathcal{J}_2}{\partial \mathbf{A}_i} + \frac{\partial \mathcal{J}_3}{\partial \mathbf{A}_i}$$

$$= \left[\frac{\partial \mathcal{J}_1}{\partial \mathbf{A}_i}\right]^+ - \left[\frac{\partial \mathcal{J}_1}{\partial \mathbf{A}_i}\right]^- + \alpha \left[\frac{\partial \mathcal{J}_2}{\partial \mathbf{A}_i}\right]^+ + \beta \left(\left[\frac{\partial \mathcal{J}_3}{\partial \mathbf{A}_i}\right]^+ - \left[\frac{\partial \mathcal{J}_3}{\partial \mathbf{A}_i}\right]^-\right), \tag{12}$$

where $\left[\frac{\partial \mathcal{J}_k}{\partial \mathbf{A}_i}\right]^+$ and $\left[\frac{\partial \mathcal{J}_k}{\partial \mathbf{A}_i}\right]^-$ are the non-negative components in the partial derivative of $\mathcal{J}_k, \forall k = 1, 2, 3$ with respect to $\mathbf{A}_i, \forall i = g, x, y, z$.

The loss function $\mathcal{J}$ will be monotonically decreased by alternatively updating each factor matrix $\mathbf{A}_i, \forall i = g, x, y, z$ using the following multiplicative updating rule in each iteration until convergence,

$$[\mathbf{A}_i]_{ab} \leftarrow [\mathbf{A}_i]_{ab} \left(\frac{\left[\frac{\partial \mathcal{J}_1}{\partial \mathbf{A}_i}\right]^-_{ab} + \left[\frac{\partial \mathcal{J}_3}{\partial \mathbf{A}_i}\right]^-_{ab}}{\left[\frac{\partial \mathcal{J}_1}{\partial \mathbf{A}_i}\right]^+_{ab} + \left[\frac{\partial \mathcal{J}_2}{\partial \mathbf{A}_i}\right]^+_{ab} + \left[\frac{\partial \mathcal{J}_3}{\partial \mathbf{A}_i}\right]^+_{ab}}\right). \tag{13}$$

## Hyperparameter tuning

To tune the hyperparameters $\alpha$ and $\beta$ in the loss function Eq (8), we applied grid search over the ranges {0, 1, 1e1, 1e2, 1e3, 1e4} and {0, 1e–3, 1e–2, 1e–1, 1, 1e1} to find the best combination that minimizes the reconstruction loss $\mathcal{J}_1$ while restricting the over-expression outside tissue mask to be less than a very small threshold such as 1e–4. This restriction can be achieved by imposing a relatively large $\alpha$ such that a small $\mathcal{J}_2$ loss is assured.

We determined the rank of CPD using $\min(\prod_i k_i, 500), \forall i = x, y, z$, where $k_i$ denotes the number of principal components explaining more than 95% variance of the RNA tomography data along different spatial axes $\mathbf{X}_i, \forall i = x, y, z$. This choice of the rank guarantees a sufficiently large rank for a good approximation under the Kruskal condition [23]. By default, CTFacTomo terminates when the sum of the differences between the previous and current components in each mode is smaller than a small threshold such as 1e–3 or 1e–4, or when the maximum iteration limit (*500* or *1000*) was reached. In both ST data and RNA tomography data, IPF [14,15] terminates either when the differences between the given and the reconstructed 1D spatial expressions in the sum over the three spatial axes are all less than a threshold *1* or upon reaching the maximum iteration limit *200*. Tomographer [16] was tuned by following the guidelines provided in the tutorial of the package, and the default settings were applied in all the experiments.

## Time and space complexity

Let $n_r = R$ and $|\cdot|$ denote the number of non-zero entries in either a matrix or a tensor, the time complexity to update $\mathbf{A}_i, \forall i = g, x, y, z$ in each iteration is $\mathcal{O}(\sum_{j \neq i} n_r |X_j| + n_r |\mathcal{M}| + \sum_j (n_r^2 n_j + n_r n_j^2))$, which is derived based on the time complexity

of matricized tensor times Khatri-Rao product (MTTKRP) [24,25]. Since in RNA tomography data, the number of genes is larger in several magnitudes, i.e., $n_i << n_g, \forall i = x, y, z, r$, the time complexity is likely to be upper bounded by either $\mathcal{O}(n_r n_g^2)$ or $\mathcal{O}(n_r n_x n_y n_z)$, in neither of which the complexity in the size of the full 4-ways of the tensor is needed. The space required for the inputs RNA tomography data, spatial and functional graphs, as well as 3D image mask is $\mathcal{O}(\sum_{i \neq g} |\mathbf{X}_i| + |\mathcal{M}| + \sum_i(|\mathbf{W}| + n_g n_r))$. In practice, CTFacTomo converges within 10 minutes and requires up to 10GB of memory for computation on the real datasets tested in this study.

## Evaluation metrics

**Reconstruction errors.** To assess the reconstruction performance of projected 3D spatial transcriptome, we applied three widely used metrics including mean square error (MSE), mean absolute error (MAE), and coefficient of determination $R^2$ over spots or genes. These metrics are defined as follows,

$$\text{MSE} = \frac{1}{n} \sum_{i=1}^{n} (\mathbf{t}_i - \hat{\mathbf{t}}_i)^2$$

$$\text{MAE} = \frac{1}{n} \sum_{i=1}^{n} \left| \mathbf{t}_i - \hat{\mathbf{t}}_i \right|$$

$$R^2 = 1 - \frac{\sum_{i=1}^{n} (\mathbf{t}_i - \hat{\mathbf{t}}_i)^2}{\sum_{i=1}^{n} (\mathbf{t}_i - \frac{1}{n} \sum_{j=1}^{n} \mathbf{t}_j)^2},$$

(14)

where $\mathbf{t} \in \mathbb{R}^n$ denotes the expression of each spot ($n = n_g$) or gene ($n = n_x \times n_y$) in the original raw spatial transcriptomics data $\mathcal{T}$ while $\hat{\mathbf{t}} \in \mathbb{R}^n$ denotes the expression of each spot ($n = n_g$) or gene ($n = n_x \times n_y$) from the imputed spatial transcriptomics data $\hat{\mathcal{T}}$ after combining the predictions of each fold in the cross-validation. Here, MSE and MAE measure the overall accuracy of the imputed values on the scale of the original data, whereas $R^2$ evaluates the proportion of variance in the ground-truth values in $\mathbf{t}$ explained by the imputed values in $\hat{\mathbf{t}}$. The formulation of $R^2$ is particularly suitable for sparse data, where correlation-based metrics may be less informative for measuring fitness of imputation tasks. Additionally, $R^2$ can be computed either across all genes within each spot (spot-wise) or across all spots for each gene (gene-wise), allowing for performance assessment that is both interpretable and comparable across genes and spots [18,19].

**Spatial coherence.** To compare slices of the CTFacTomo and IPF reconstructions on RNA tomography data with data coming from matched slices of the Stereo-seq data, we used a bivariate version of Moran's I score [26], of which the univariate version was first introduced in [27]. The bivariate formulation is shown below.

$$\text{Moran score } (\mathbf{x}, \mathbf{y}) = \frac{\sum_i \sum_j \mathbf{w}_{ij}(\mathbf{x}_i - \bar{\mathbf{x}})(\mathbf{y}_j - \bar{\mathbf{y}})}{\sqrt{\sum_i (\mathbf{x}_i - \bar{\mathbf{x}})^2} \sqrt{\sum_j (\mathbf{y}_j - \bar{\mathbf{y}})^2}}$$

$$\mathbf{w}_{ij}^0 = \exp\left(-\frac{\mathbf{d}_{ij}^2}{2l^2}\right); \ \mathbf{w}_{ij} = \frac{n}{W} \mathbf{w}_{ij}^{(0)}.$$

(15)

Here, $\mathbf{x}_i$ and $\mathbf{y}_j$ are the gene expression of a gene at the i-th spot in the reconstructed slice and the expression at the j-th spot in the Stereo-seq slice. Accordingly, $\bar{\mathbf{x}}$ and $\bar{\mathbf{y}}$ are the mean expressions of the gene in the slice. $\mathbf{w}_{ij}^{(0)}$ is the unnormalized weight between the spatial location of spot i and spot j. We then normalize using the number of spots $n$ and $W$, the sum of the weight matrix. Here, $n$ refers to the number of Stereo-seq spots which is different for every Stereo-seq slice. One important difference to note as compared to more common previous uses of the bivariate Moran score is that the number of spots indexed by $i$ and the number of spots indexed by $j$ are different for the use case. This is necessary since the grid of the spots in the Stereo-seq slice and the reconstruction from Tomo-seq data have different arrangements

depending on the resolution of the data. In addition, other than calculating the score between all the spots from the compared slices, we also calculated the Moran score only between the top spots. Specifically, we set $\mathbf{w}_{ij} = 0$, if $\mathbf{y}_j = 0$ in the Stereo-seq slice or $\mathbf{x}_i$ is in the top $2n_{>0}$ spots in the reconstructed slice, where $n_{>0}$ is the number of non-zero spots in $\mathbf{y}_j$. This variation allows the score to focus on non-zero (or top) entries for a more sensitive evaluation of the sparse data.

**Enrichment significance.** To measure the 3D reconstruction performance with Tomo-seq data for gene clustering, we computed the log of the $q$-value of the most significant enriched GO term for each gene cluster and then averaged these minimal q-values across all gene clusters to evaluate the overall enrichment significance. We performed enrichment over $10,185$ GO terms from the C5 collection in the Molecular Signatures Database (MSigDB), which includes $7,751$ biological process (BP) terms, $1,009$ cellular component (CC) terms, and $1,772$ molecular function terms. We calculated q-values by adjusting enrichment $p$-values by false discovery control (FDR) with the Benjamini-Hochberg (BH) procedure.

### Alignment of Stereo-seq and reconstructed RNA tomography slices

To use the slice-wise Stereo-seq zebrafish embryo data as a validation dataset to further assess reconstruction performance, twelve middle slices of the reconstructed gene expression data needed to be aligned with the twelve available Stereo-seq slices. Note, the reconstructed slices are $18\mu m$ and the Stereo-seq slices are $12\mu m$ in thickness. We first normalized the gene expression of the Stereo-seq data and the reconstructed gene expression data by dividing by the magnitude of the gene expression matrix. We then used PASTE's pairwise_align function with default values for the parameters to align the slices after normalization [28]. The output of the function is a list of the input slices–two in our case–with the coordinates of their spots adjusted so as to be more aligned with each other. We then use these aligned slices as input to the Moran score calculation.

### Compared methods

CTFacTomo is benchmarked against the two available best methods, IPF algorithm [14,15] and probabilistic graphical model Tomographer [16].

**IPF** is a simple scaling-based procedure for estimating the fitted tensor, $\hat{\mathcal{T}}$, from the given marginals of the target tensor, $\mathcal{T}$, which alternately minimizes the discrepancies between the marginals of $\mathcal{T}$ and $\hat{\mathcal{T}}$ along different axes until convergence. In this study, we followed the MATLAB implementation described in [14,29] for 3D spatial transcriptome reconstruction with Tomo-seq [14] and rewrote the IPF algorithm in Python.

**Tomographer** is a compressed sensing algorithm to reconstruct the matrix from sampled marginal distributions by maximizing the posterior of their joint distribution. It is specifically designed for 2D spatial transcriptome reconstruction with STRP-seq [16], an RNA tomography technique that sections the tissue into consecutive slices and further cuts slices into strips at different angles. For 3D transcriptome reconstruction, Tomographer can be adapted by reconstructing a 2D spatial transcriptome for each tissue slice at a specific location along the z-axis at a time and subsequently stacking these reconstructions. In this study, we used the Tomographer package provided in the work [16] for our experiments.

Given Tomographer was designed for 2D STRP-seq data, in our comparisons, we applied Tomographer to 3D spatial transcriptomics datasets using simulated 2D slices generated from projections at 0˚ and 90˚ in each slice. This approach is not applicable to real 3D Tomo-seq datasets, as ground-truth spatial gene expression is unavailable and the Tomo-seq projections are inherently three-dimensional such that the x- and y-axis projections do not correspond to any individual 2D slice. Note that there is also no 3D STRP-seq data for evaluation. Thus, the comparison to Tomographer is only conducted in the projected 3D ST datasets.

### Results

Three experiments were conducted for the evaluation of CTFacTomo. The first experiment focused on three pseudo RNA tomography datasets constructed from projections of 3D spatial transcriptomics data, including two low-resolution

ST1K datasets: a mouse brain tissue with *40* slices [30] and a human heart tissue with *9* slices [31], and a high-resolution Stereo-seq dataset from a Drosophila embryo with *21* slices [32]. The second experiment was conducted on two real RNA tomography datasets, one from a zebrafish embryo ((50, 49, 56) slices) [14] and the other from a mouse olfactory mucosa ((56, 54, 60) slices) [29]. Finally, for both qualitative and quantitative evaluation, the reconstructed data of the zebrafish embryo was also compared with a 3D Stereo-seq dataset with the matched stage and tissue region [33]. All these datasets are summarized in Table 2.

## Data preparation

The human heart ST dataset contains a stack of *9* adjacent slices of 2D ST spatial transcriptomics data from a developing human heart at 6.5 post-conception weeks (PCW) sectioned along the ventrodorsal axis. The mouse brain ST dataset contains a stack of *40* slices sectioned along the anteroposterior axis of an adult mouse brain. In both datasets, each ST slice contains $33 \times 35$ spots, and these slices were manually registered based on their associated H&E staining images and batch-corrected across slices [30,31]. The Drosophila embryo 3D spatial transcriptomics dataset consists of a stack of *21* slices sectioned along the left-right axis of the Drosophila embryo at the second instar larva (L2) stage [32]. All three datasets were normalized in log-space.

In the two ST datasets, we binned the spots in the stack of 2D expression data into a 3D tensor to reconcile the coordinate shift across the slices. The spots were assigned to the closest bin and averaged within the bin. The dimensions after binning are $14,726 \times 20 \times 20 \times 9$ for the human heart and $10,239 \times 30 \times 30 \times 30$ for the mouse brain, after removing non-expressed genes and background regions for both tissues. The Drosophila embryo data was directly formed into a $14,270 \times 124 \times 105 \times 21$ tensor, with non-expressed genes and background regions being removed, and then binned into a $14,270 \times 63 \times 53 \times 21$ tensor to amplify the expression signals in simulation studies. Each slice contains $124 \times 105$ spots, each of which is a bin of $50 \times 50$ DNA Nanoballs (DNBs) as processed in the original study. The 3D mask was derived for the no-expression spots by collapsing gene expressions along the gene axis in the tensor. A spatial chain graph was constructed by the number of slices in each spatial axis and the protein-protein interaction (PPI) networks for *Homo sapiens*, *Mus musculus*, and *Drosophila melanogaster* were obtained from BioGRID version 4.4.242 [34] using mainly only physical interactions.

In the second experiment, we evaluated 3D expression reconstruction on real RNA tomography data. The first dataset contains (50, 49, 56) slices along three spatial axes from a zebrafish embryo at the shield stage [14]. The second dataset contains (56, 54, 60) slices in the three views from a mouse olfactory mucosa [29]. We constructed a *Danio rerio* PPI network from STRING [35] with the interactions of confidence scores at least 0.8 retained for sufficient connectivity in the network. The *Mus musculus* PPI network was also constructed from BioGRID [34]. After the intersection of the PPI network and the gene expression profiles, 6, 153 and 9, 254 genes are retained in the zebrafish embryo and the mouse olfactory mucosa, respectively.

**Table 2. Summary and specifications of the datasets.**

| Experiment | Tissue | Platform | Original Size (genes,x,y,z) | Reconstructed Size (genes,x,y,z) |
|---|---|---|---|---|
| **3D Projection Validation** | Mouse adult brain | ST1K | $10,239 \times 35 \times 33 \times 30$ | $10,239 \times 30 \times 30 \times 30$ |
| | Developing human heart | ST1K | $14,726 \times 35 \times 33 \times 9$ | $14,726 \times 20 \times 20 \times 9$ |
| | Drosophila embryo | Stereo-seq | $14,270 \times 124 \times 105 \times 21$ | $14,270 \times 63 \times 53 \times 21$ |
| **Tomography Reconstruction** | Mouse olfactory mucosa | Tomo-seq | $6,153 \times (50 + 49 + 56)$ | $6,153 \times 50 \times 49 \times 56$ |
| | Zebrafish Embryo | Tomo-seq | $9,254 \times (56 + 54 + 60)$ | $9,254 \times 56 \times 54 \times 60$ |
| **External Evaluation** | Zebrafish Embryo | Stereo-seq | $5,934 \times 6064(xy) \times 12$ | – |

In the zebrafish embryo data, we followed the same procedure in the original study to normalize each 1D spatial expression data based on embryonic geometry derived from the image mask along the corresponding spatial axis and then log-transformed the normalized data [14]. In the mouse olfactory mucosa data, we used the strategy described in [29] to initially fit 1D expression data for each gene along different spatial axes with polynomial fitting and then normalized them with the same procedure used for zebrafish embryo data.

For validation of the reconstructed zebrafish embryo data, we used an additional external Stereo-seq dataset that provides a spatiotemporal mapping of the zebrafish development that provides a more granular, ground-truth slice-wise view of gene expression [33]. We specifically used the embryo data at 5.25 hours post-fertilization (hpf) from this dataset to compare with our reconstruction on the Tomo-seq data, which was in the shield stage (6 hpf). Though at slightly different time points, both are in the gastrula period of zebrafish embryo development [36].

**CTFacTomo accurately reconstructs 3D spatial gene expressions from projected 3D spatial transcriptomics data**

The task in this experiment is to reconstruct the ground-truth 3D expressions based on the tomographically aggregated gene expressions from the three curated 3D spatial transcriptomics datasets listed in Table 2. The spatial gene expressions in the 3D datasets were regarded as ground truth, and the 3D expressions were projected into three 1D spatial gene expression matrices that resemble RNA tomography data along the three orthogonal spatial x-y-z axes, each matching a dimension of the 3D spatial expression tensor.

In this experiment on the projected 3D ST data, hyperparameters $\{\alpha, \beta, \text{rank}\}$ were set as $\{1e2, 1e–3, 120\}$ for the mouse brain dataset, $\{1e4, 1e–2, 96\}$ for human heart dataset and $\{1e2, 1e–3, 120\}$ for drosohpila embryo data for reconstruction using the parameter tuning strategy described in the **Methods** section. The complete results for hyperparameter tuning are shown in Fig 2A–2C. The reconstruction performance on the projected ST data was evaluated by mean squared error (MSE) and mean absolute error (MAE) between the reconstructed expression tensor and the ground-truth ST expressions. We also measured both spot-wise and gene-wise $R^2$ to assess how well the reconstruction recovers spot and gene patterns. As shown in Table 3, CTFacTomo consistently outperformed both IPF and Tomographer in 3D reconstruction with the lowest MSE and MAE and the highest spot-wise and gene-wise $R^2$ on all three datasets. Note that IPF does not work in the high sparsity of the Drosophila embryo data, and thus is not applicable in the comparison.

For understanding the variation in $R^2$, the distributions of spot-wise and gene-wise $R^2$ of CTFacTomo over spot or gene density are shown in S1 Fig. It is clear the spot-wise $R^2$ positively correlates with the spot densities in all the three datasets. Majority of the extremely low $R^2$ occur in the spots of very low densities as expected. The distribution of the gene-wise $R^2$ follows a more complex pattern. At the extremely low density such as <5% on the mouse brain and Drosophila embryo datasets and <1% on the human heart dataset, the gene-wise $R^2$ is out-of-scale and thus, is too low to be evaluated in the comparisons. In the normal range of gene expression, the distribution of $R^2$ is stable and similar. Interestingly, at the extremely high density (>95%), the $R^2$ starts to decrease significantly again. These genes are typically housekeeping genes that are ubiquitously expressed, which likely do not adopt any spatial patterns and thus are more difficult to reconstruct with RNA tomography data and low-rank formulation. Thus, only the genes in the normal expression range (1%-95% on human heart dataset and 5%-95% on mouse brain and Drosophila embryo datasets) are considered in calculation of the gene-wise $R^2$ shown in Table 3. More detailed comparisons of the performance are also shown by scatter plots of $R^2$ in S2 Fig, S3 Fig and S4 Fig for the three datasets.

To further investigate the contribution of spatial and functional information in the reconstruction, we performed an ablation study by removing the corresponding graph Laplacian regularization in the total loss function ("CTFactTomo w/o graph" in Table 3). We observed that CTFacTomo outperformed its counterpart without graph Laplacian regularization, which suggests that both spatial and functional relations are crucial guidance for the reconstruction from 1D data. In particular, in the human heart data and the Drosophila embryo data, CTFacTomo without graph regularization performed worse than IPF and Tomographer, suggesting that regularization is often necessary and required for high-order modeling

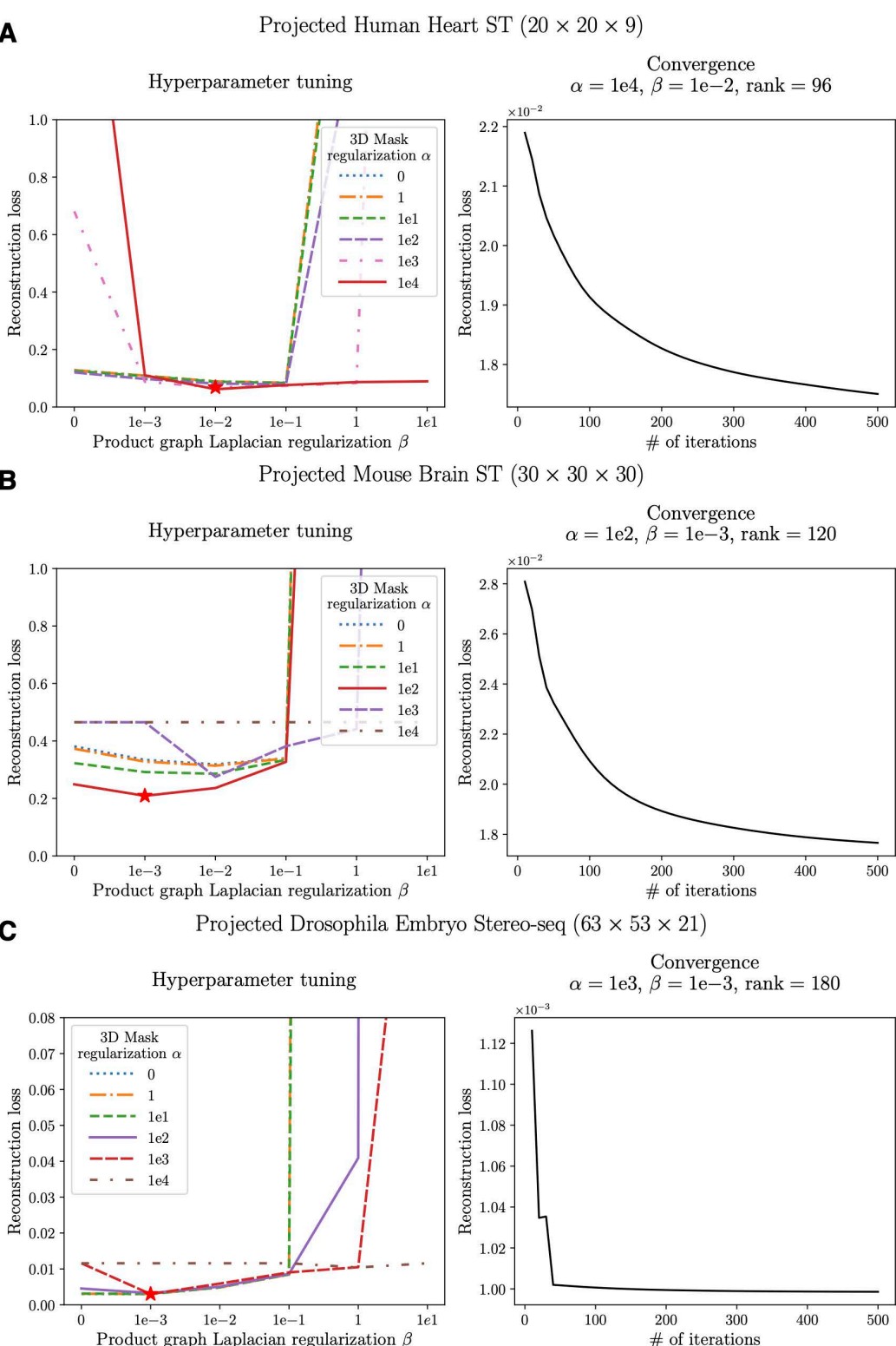

**Fig 2. Convergence and hyperparameter tuning for CTFacTomo on 3D-projected ST data. A**: Mouse brain **B**: Human heart **C**: Drosophila embryo. *left*: Hyperparameters $\alpha$ and $\beta$ are searched over the grid of {0, 1, 1e1, 1e2, 1e3, 1e4} and {0, 1e–3, 1e–2, 1e–1, 1, 1e1}; ***right***: Reconstruction loss of CTFacTomo with optimal hyperparameters decreases over the iterations. The optimal hyperparameters (marked by red stars) are determined by the combination that minimizes the reconstruction loss while restricting the expression outside the mask.

**Table 3. Reconstruction performance on 3D-projected spatial transcriptomics data.** The comparison of IPF, CTFacTomo, CTFacTomo without graph Laplacian regularization, and Tomographer on human heart, mouse brain, and Drosophila embryo data. The best mean square error (MSE), mean absolute error (MAE), spot- and gene-wise $R^2$ are bold.

| Dataset | Method | Evaluation Metrics | | | |
|---|---|---|---|---|---|
| | | *MSE ↓* | *MAE ↓* | *spot-wise $R^2$ ↑* | *gene-wise $R^2$ ↑* |
| **Human Heart** $(14,726 \times 20 \times 20 \times 9)$ | IPF | $0.160 \pm 0.081$ | $0.190 \pm 0.061$ | $0.285 \pm 0.205$ | $0.039 \pm 0.120$ |
| | Tomographer | $0.158 \pm 0.089$ | $0.170 \pm 0.056$ | $0.337 \pm 0.164$ | $0.059 \pm 0.183$ |
| | CTFacTomo w/o graph reg | $0.176 \pm 0.163$ | $0.217 \pm 0.192$ | $0.269 \pm 0.230$ | $0.011 \pm 0.259$ |
| | CTFacTomo w/ graph reg | $\mathbf{0.147 \pm 0.093}$ | $\mathbf{0.156 \pm 0.055}$ | $\mathbf{0.386 \pm 0.181}$ | $\mathbf{0.120 \pm 0.148}$ |
| **Mouse Brain** $(10,239 \times 30 \times 30 \times 30)$ | IPF | $0.426 \pm 0.159$ | $0.389 \pm 0.081$ | $0.262 \pm 0.245$ | $0.011 \pm 0.188$ |
| | Tomographer | $0.293 \pm 0.136$ | $0.363 \pm 0.080$ | $0.472 \pm 0.183$ | $0.024 \pm 0.121$ |
| | CTFacTomo w/o graph reg | $0.288 \pm 0.196$ | $0.321 \pm 0.104$ | $0.517 \pm 0.289$ | $0.096 \pm 0.193$ |
| | CTFacTomo w/ graph reg | $\mathbf{0.245 \pm 0.172}$ | $\mathbf{0.298 \pm 0.097}$ | $\mathbf{0.600 \pm 0.174}$ | $\mathbf{0.143 \pm 0.083}$ |
| **Drosophila Embryo** $(14,270 \times 63 \times 53 \times 21)$ | IPF | − | − | − | − |
| | Tomographer | $0.017 \pm 0.227$ | $0.020 \pm 0.130$ | $0.173 \pm 0.187$ | $0.089 \pm 0.083$ |
| | CTFacTomo w/o graph reg | $0.019 \pm 0.245$ | $0.021 \pm 0.137$ | $0.075 \pm 0.009$ | $0.040 \pm 0.110$ |
| | CTFacTomo w/ graph reg | $\mathbf{0.010 \pm 0.140}$ | $\mathbf{0.017 \pm 0.100}$ | $\mathbf{0.449 \pm 0.096}$ | $\mathbf{0.169 \pm 0.070}$ |

used in CTFacTomo. Note that while Tomographer appears to perform better than IPF with the two-angle (0° and 90°) setting, STRP-seq requires dissecting tissues along the z-axis first and then, further sectioning each slice into parallel strips along either the x-axis (90°) or y-axis (0°), which makes tissue preparation for STRP-seq significantly more complex compared to Tomo-seq in practice. Note that PPI networks are typically scale-free, in which the degree distribution plays a critical role and is often closely related to the leading eigenvector of the graph Laplacian. Consequently, when the PPI network is randomized but preserves the degree distribution, the retained structural information can still provide benefits for downstream tasks such as imputation, clustering or classification [18,19]. In the comparisons, the differences between the original and randomized networks depend on the weight assigned to the graph regularization term, although the contribution of PPI information remains evident.

## CTFacTomo reconstructs 3D spatial expressions with consistent patterns from RNA tomography data

In this experiment, we evaluated 3D expression reconstruction on real RNA tomography data. The first dataset contains (50, 49, 56) slices along three spatial axes from a zebrafish embryo at the shield stage [14]. The second dataset contains (56, 54, 60) slices in the three views from a mouse olfactory mucosa [29] as listed in Table 2.

**3D transcriptome reconstruction from RNA tomography for zebrafish embryo.** For the RNA tomography data of zebrafish embryo, the hyperparameters $\alpha = 1, \beta = 1, \text{rank} = 500$ were chosen for this dataset based on the hyperparameter tuning results shown in Fig 3A. The reconstruction loss under different ranks is also shown in S1 FigA, which clearly demonstrate that the determined rank is low yet sufficient for the reconstruction task.

To compare the reconstructions of CTFacTomo and IPF from RNA tomography data, we performed a quantitative slice-wise comparison of the reconstructions by CTFacTomo and IPF with the slices in the external Stereo-seq dataset [33].

All the slices for comparison are along the sagittal plane. The original Tomo-seq slices have a thickness of $18\mu m$ and the Stereo-seq slices have a thickness of $12\mu m$. We used a bivariate variation of Moran's I score as a quantitative measure of spatial correlation to compare a pair of reconstructed and Stereo-seq slices (see **Methods** section). We took the twelve middle slices of the reconstructed zebrafish embryo and compared them to the twelve available slices of Stereo-seq data that were obtained from the matched middle sections of the embryo. For each gene, we obtained 144 Moran scores, one for each of the pairs. We then consider the maximum score to pick matched reconstructed and Stereo-seq slices accordingly for several comparisons and visualizations.

First, for each gene in each Stereo-seq slice, we selected the reconstructed slice of the same gene generated by either CTFacTomo or IPF that best matched it. We then show the scatter plots between the Moran scores of the top 2,000 highly variable genes when comparing the best reconstruction by CTFacTomo and IPF against the 2,000 genes in each Stereo-seq slice in Fig 4. It is clear that CTFacTomo provided significantly better reconstruction of the genes in all the 12 Stereo-seq slices. To further investigate the implications of the spatial difference in the Moran scores, we selected eighteen genes with the maximum differences in the scores in the scatter plots (marked in red circles), and manually examined their spatial patterns to provide explanations for validating the results. The expression of these genes and the explanations are shown in S9 Fig–S26 Fig. All the cases show distinguishable differences in matching the spatial patterns that are consistent with the difference in Moran scores. In a second comparison in Fig 5, we also show the global analysis of the Moran scores for all 5,439 genes and the top 2,000 highly variable genes matched between Tomo-seq data and Stereo-seq data. In this comparison, the best Moran score is selected between each gene in the reconstruction by CTFacTomo or IPF and the 12 Stereo-seq slices, and thus measure the performance of all the reconstructed data. The results also show that CTFactTomo reconstructed significantly more consistent spatial gene expressions than IPF in both the top 2,000 highly variable genes and all 5,439 genes. This conclusion was consistent in both the case of when only the top spots were used to calculate the Moran score and also when all the spots were used.

We further evaluated the reconstruction performance by visualizing the expression of several well-known marker genes and comparing their 2D projections with existing ISH images from the same perspective [14]. ISH images are available for four genes, *GSC*, *SULF1*, *ISM1*, and *MAGI1B*, and their reconstruction are shown in Fig 6A. All of these four genes are expressed as a patch or different gradients in the organism on the dorsal side at the early development stages of the zebrafish embryo. *GSC* is an important gene in developing mesoderm. While the 3D reconstruction of both CTFacTomo and IPF in some projections, such as along the frontal and anteroposterior axes, highlight the correctly expressed regions in the ISH images, CTFacTomo reconstructions align better with the ISH images without overspreading expression. Moreover, it is obvious that the projections of 3D reconstructions from CTFacTomo along the ventrodorsal axis demonstrated significantly higher agreement with their ISH images compared to IPF.

The best Moran scores of the matched slices for the five marker genes are reported in Table 4 for comparing CTFacTomo and IPF. The detailed scores are shown in S6 Fig. It is clear that the reconstructed slices by CTFacTomo matches the spatial pattern in the Stereo-seq slices better with higher Moran scores for all the genes. In Fig 6B, we visualize the pairs with the best Moran score for five marker genes (*GSC*, *MAGI1B*, *MESPAB*, *NET1*, *INSB*). In the plots, we see that the *GSC* reconstruction from CTFacTomo better corresponds with the Stereo-seq data than IPF. The right region should be the most highly expressed for this slice as suggested by Stereo-seq, a pattern that CTFacTomo captures. On the other hand, IPF places high levels of expression on both right and left regions of the slice but in fact the left region shows no expression at all. While CTFacTomo also incorrectly places spots in that left region, those spots are relatively low expressions in comparison to the rest of the slice. Upon visualization, it is also clear that IPF often tends to overspread expression. For example, while both the CTFacTomo and IPF reconstructions of *MAGI1B* generally capture the left, middle, and right regions of expression as supported by Stereo-seq, IPF misses that the center region should be more highly expressed as it has relatively high expression all throughout the slice. For *MESPAB*, CTFacTomo and IPF perform similarly but again overspreading can be seen in that IPF still has relatively high expression throughout the slice even in

# A

## Zebrafish Embryo (Shield Stage) (54 × 49 × 56)

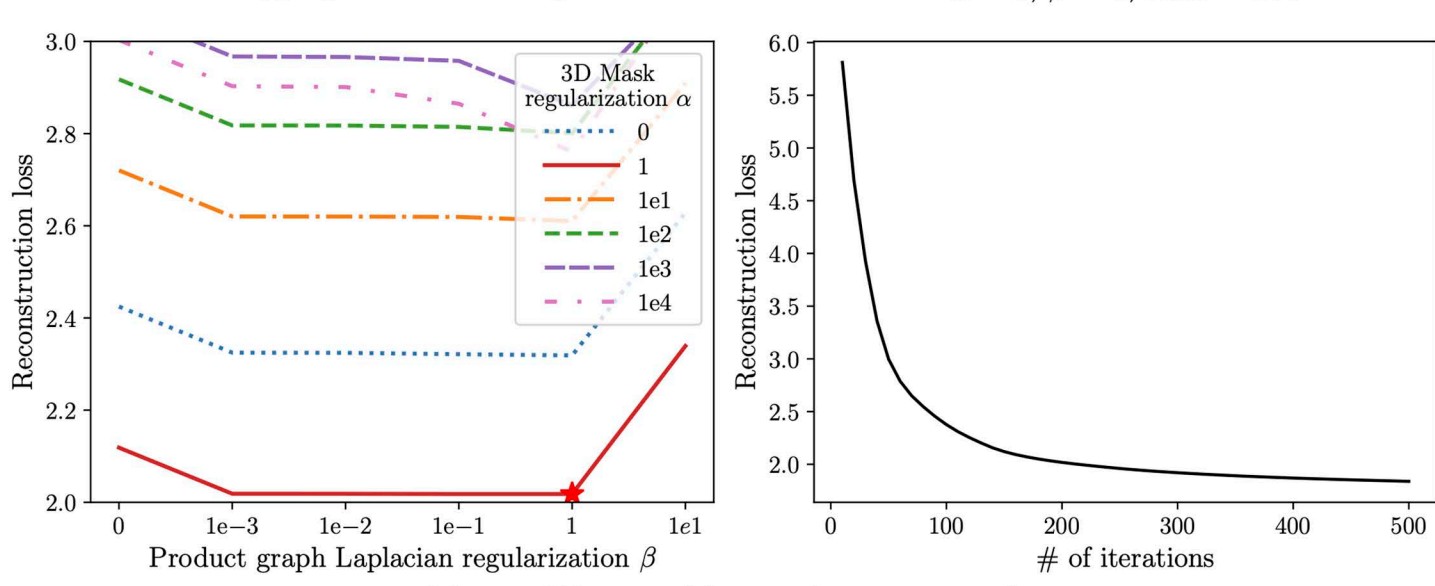

# B

## Mouse Olfactory Mucosa (56 × 54 × 60)

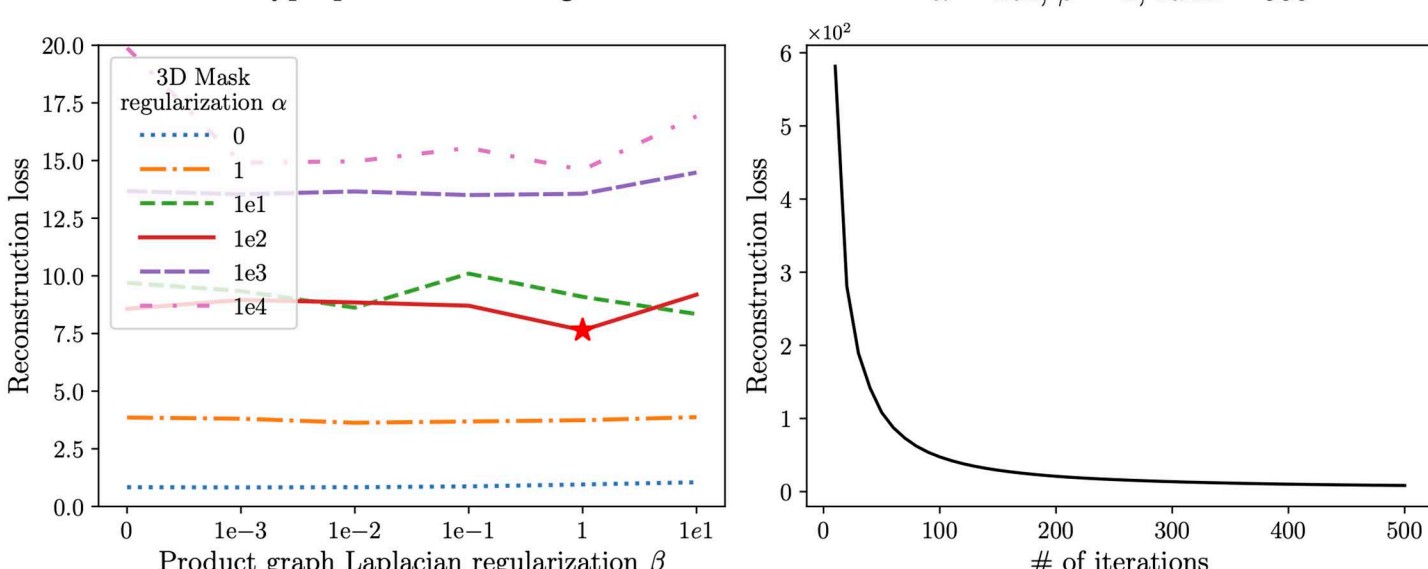

**Fig 3. Convergence and hyperparameter tuning of CTFacTomo on RNA tomography data. A**: Zebrafish embryo; and **B**: Mouse olfactory mucosa. *left*: Hyperparameters $\alpha$ and $\beta$ are searched over the grid of {0, 1, 1e1, 1e2, 1e3, 1e4} and {0, 1e−3, 1e−2, 1e−1, 1, 1e1}; *right*: Reconstruction loss of CTFacTomo with optimal hyperparameters decreases over the iterations. The optimal hyperparameters are determined by the combination that minimizes the reconstruction loss while restricting the expression outside the tissue mask to be less than threshold 1e−4. Note that in the experiment on mouse olfactory mucosa data, $\alpha$ needs to be at least 1e2 to restrict the level of expression outside the tissue mask to be less than threshold 1e−4. Thus, $\alpha$ =1e2 was selected as the best parameter in the tuning.

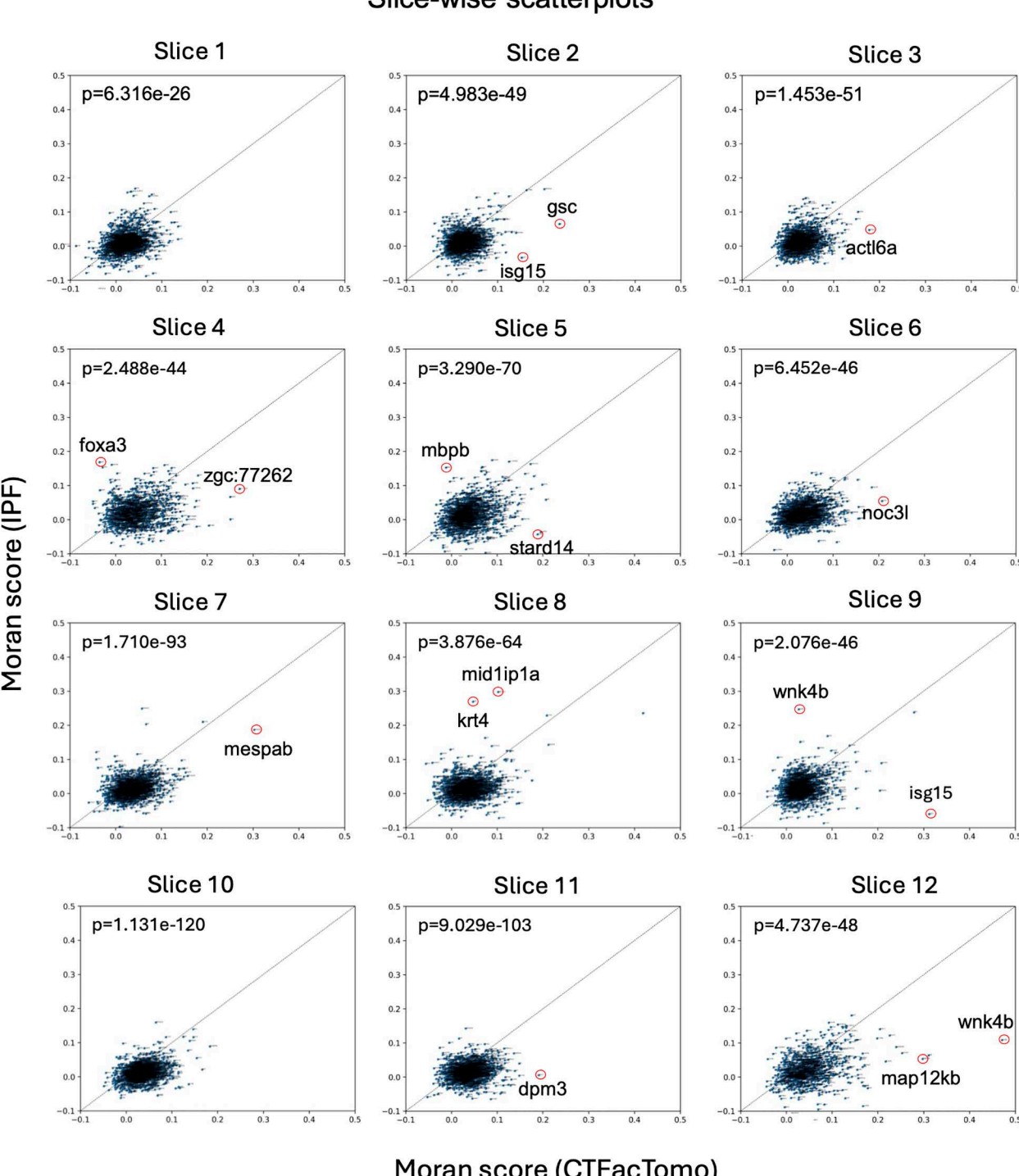

**Fig 4. Comparison of spatial coherence between reconstructed and Stereo-seq slices by Moran scores.** The scatter plots compare the Moran scores of the top 2,000 highly variable genes when comparing the expressions in a Stereo-seq slice to the best matched reconstruction by CTFacTomo and IPF. *P*-values by *t*-test are also shown for each slice. The genes with the maximum differences in the scores in the scatter are marked in red circles.

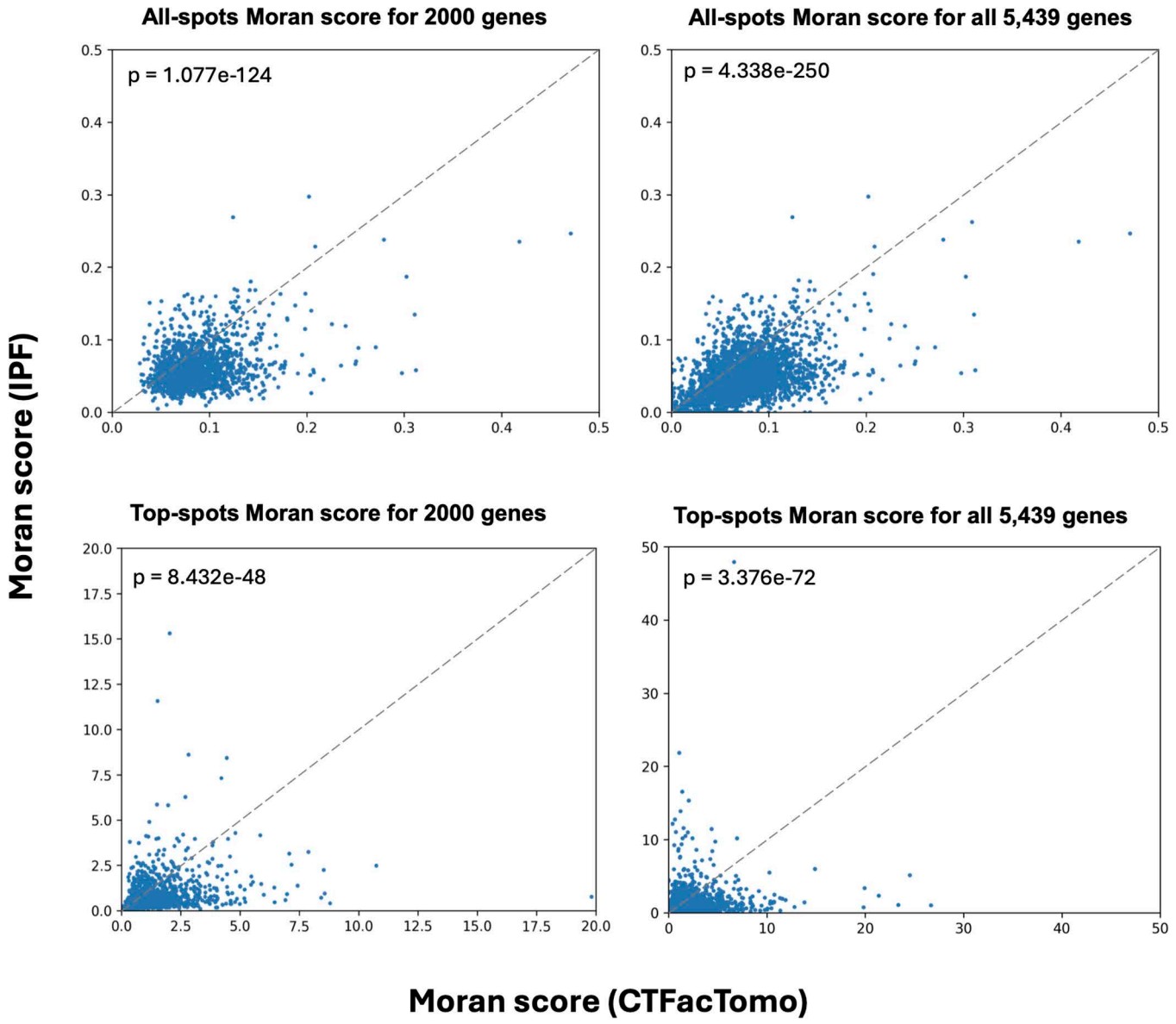

**Fig 5. Moran scores of all genes in Zebrafish embryo data; Comparison of spatial coherence between reconstructed and Stereo-seq slices by Moran scores.** The scatter plots compare the Moran scores of the top 2,000 highly variable genes or all 5,439 genes when comparing the reconstructed data by CTFacTomo and IPF to Stereo-seq slice-wise data. The first row shows the Moran scores calculated over all spots and the second row shows the Moran scores calculated over the top spots.

regions not represented in the Stereo-seq. The reconstruction of *NET1* by both IPF and CTFacTomo tend to over-expressed in all regions but CTFacTomo has less over-dispersion. The reconstruction of *ISNB* by CTFacTomo apparently captures the middle region and the right-end region more precisely. These comparisons again confirm the better matching of the reconstruction by CTFacTomo than IPF.

**3D Transcriptome reconstruction from RNA tomography for mouse olfactory mucosa.** In the experiment on the RNA tomography dataset of mouse olfactory mucosa, the hyperparameters $\alpha = 100$, $\beta = 1$, rank = 500 were selected

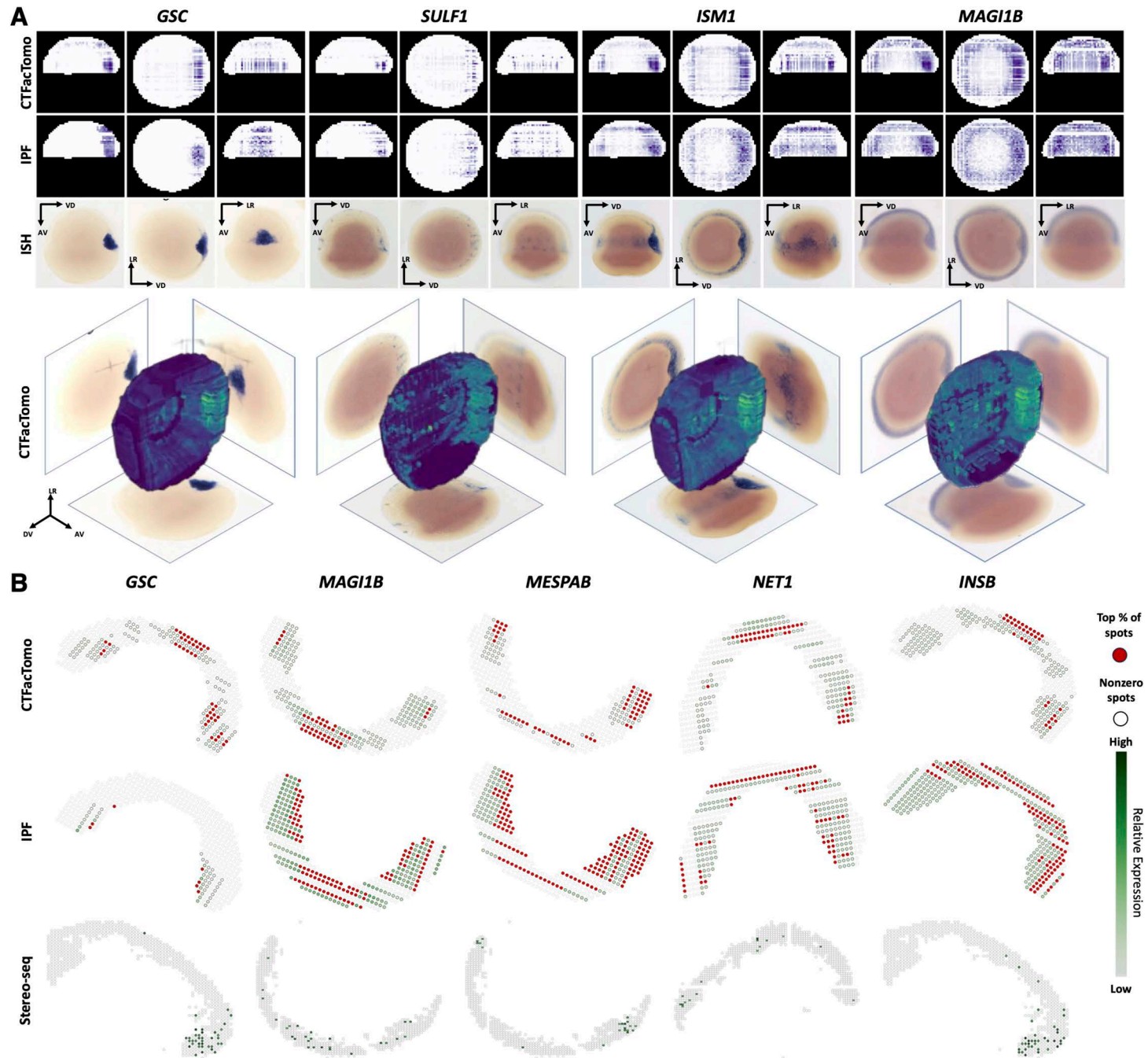

**Fig 6. Visualization of reconstructed 3D expression for marker genes; Visualization and evaluation of marker genes in zebrafish Tomo-seq data. A**: 2D projection of the reconstructed expression for the entire zebrafish embryo at shield stage along frontal, anteroposterior, and ventrodorsal axes of four genes (*GSC*, *SULF1*, *ISM1*, and *MAGI1B*). Shaded blue areas in the ISH images denote expression regions. The bottom row shows the 3D visualization of the reconstruction by CTFacTomo. **B**: A comparison between the reconstructed slices by CTFacTomo and IPF with the best matched Stereo-seq slices by Moran score for five marker genes (*GSC, MAGI1B, MESPAB, NET1, INSB*). In all the plots except those for *MESPAB*, spots with a black outer edge represent non-zero expression > 0.0001, and top spots that cover 50% of the total expression are colored red to highlight highly expressed regions. The reconstructed expression for *MESPAB* by IPF is very close to uniform, thus non-zero expression threshold is set to > 0.00001 and the top spots that cover 75% of expression are shown in red in its plots.

**Table 4. Moran scores between the reconstructed data and the matched Stereo-seq data for five marker genes. All spots or top spots are used in the calculation.**

| Marker Gene | All spots | | Top spots | |
|---|---|---|---|---|
| | IPF | CTFacTomo | IPF | CTFacTomo |
| *GSC* | 0.0644 | **0.2350** | 0.4621 | **2.3425** |
| *MAGI1B* | 0.0619 | **0.1528** | 0.3596 | **1.0123** |
| *MESPAB* | 0.1870 | **0.3022** | 1.3065 | **2.9081** |
| *NET1* | 0.0473 | **0.0966** | 0.6298 | **1.2809** |
| *INSB* | 0.0744 | **0.0885** | 0.4669 | **0.7230** |

based on the hyperparameter tuning results shown in Fig 3B. The reconstruction loss under different ranks is also shown in S1 FigB, which clearly demonstrate that the determined rank is low yet sufficient for the reconstruction task. The stopping criteria was to have a residual threshold of 1e–4 and a max epoch of 1000. Similar to the previous experiment, we visualized the reconstructions from CTFacTomo ($9,254 \times 56 \times 54 \times 60$) and IPF for comparisons on five marker genes with ISH images available [29]. The five marker genes (*OLFR309*, *OLFR618*, *OLFR727*, *CYTL1*, *MOXD2*) are shown in Fig 7. *OLFR309*, *OLFR727*, and *OLFR618* are three olfactory receptors and genetic markers of different mature olfactory sensory neuron subtypes. *CYTL1* is a cytokine-like protein playing roles in osteogenesis, chondrogenesis, and bone and cartilage homeostasis [37]. *MOXD2* is a mono-oxygenase dopamine hydroxylase-like protein functioning in olfaction [38].

It is most clear when comparing a reconstructed slice of *OLFR618* along the anteroposterior axis with its last ISH image, where CTFacTomo accurately reconstructs expression, whereas IPF fails to reconstruct any. The fourth image of *OLFR727* shows another example of this, where IPF exaggerates the expression in the bottom two corners and CTFac-Tomo more accurately shows the two concentrated regions of expression. We see a similar example in the fifth image of *OLFR727*, where IPF even incorrectly infers expression in the upper region, whereas CTFacTomo correctly infers that there is only expression in the bottom region of that slice. However, there are some instances among the visualized genes in which both CTFacTomo and IPF incorrectly represent spatial gene expression. For example, both CTFacTomo and IPF struggle to accurately reconstruct the expression of *Olfr309*. Note that since *OLFR618*'s expression level is only around 1% of the other genes, we set $\alpha = 1e–3$ to run CTFacTomo separately such that the extremely low expressions can be retained in the reconstruction. The expression of all the slices are also visualized in S7 Fig for reconstructions by CTFac-Tomo and S8 Fig for reconstructions by IPF. In summary, through a visual inspection of the reconstructions, we see that CTFacTomo can find more focused expression regions than IPF, likely due to IPF's tendency to overspread the regions on such data.

### Reconstructed 3D transcriptomes from CTFacTomo enhances biological interpretation of spatially co-expressed gene clusters

To evaluate whether the reconstructed 3D spatial expressions can enhance biological interpretation, we further investigated the reconstruction on real RNA tomography data following the study design in [39]. We first applied kMeans to group genes into *100* co-expressed clusters with either concatenated 1D expressions in the original RNA tomography profiles or reconstructed 3D expressions. Gene Ontology (GO) enrichment analysis was performed by using the enrichGO function from R package clusterProfiler for each cluster, with significance determined by the *p*-value of the most significant GO term of each cluster. The comparison of the number of significant clusters by different reconstruction methods is shown in Fig 8. We observed that the 3D expressions reconstructed by CTFacTomo consistently identified more significant co-expressed gene clusters across all *p*-value thresholds for both zebrafish embryo and mouse olfactory mucosa than the data reconstructed by IPF and the original RNA

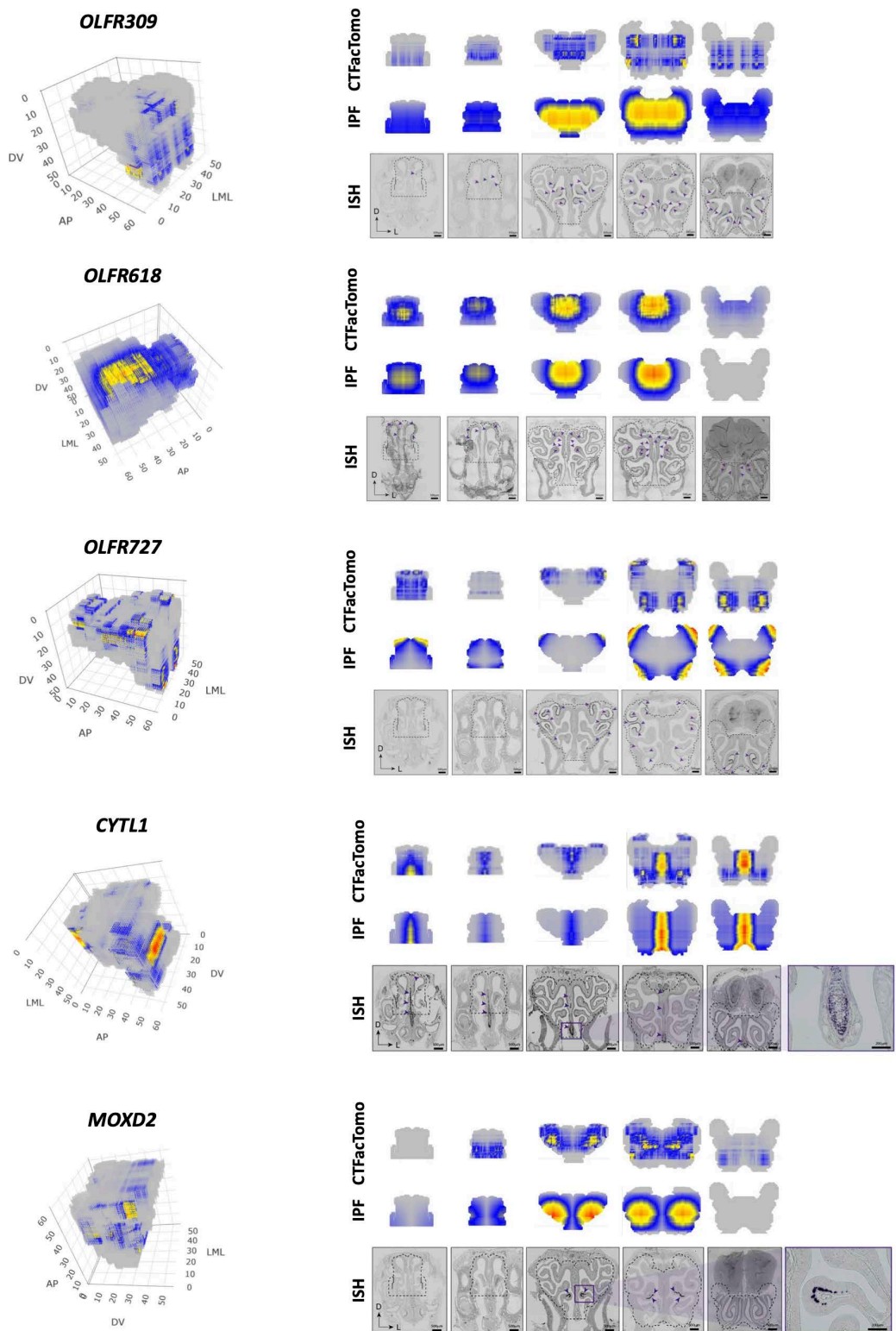

**Fig 7. Visualization of reconstructed 3D expression for marker genes; Visualization and evaluation of marker genes in mouse olfactory mucosa.** We visualize both the full 3D reconstruction as well as slice-wise visualizations for five marker genes (*OLFR309*, *OLFR618*, *OLFR727*, *CYTL1*,

*MOXD2*). **left**: The 3D gene expression output by CTFacTomo for the five marker genes. **right**: A comparison between CTFacTomo and IPF in a slice-wise view of the 3D reconstruction in comparison to the ground-truth ISH images for each gene. Within the ISH images, purple triangles represent regions of expression. The fullsized ISH images are shown in S27 Fig.

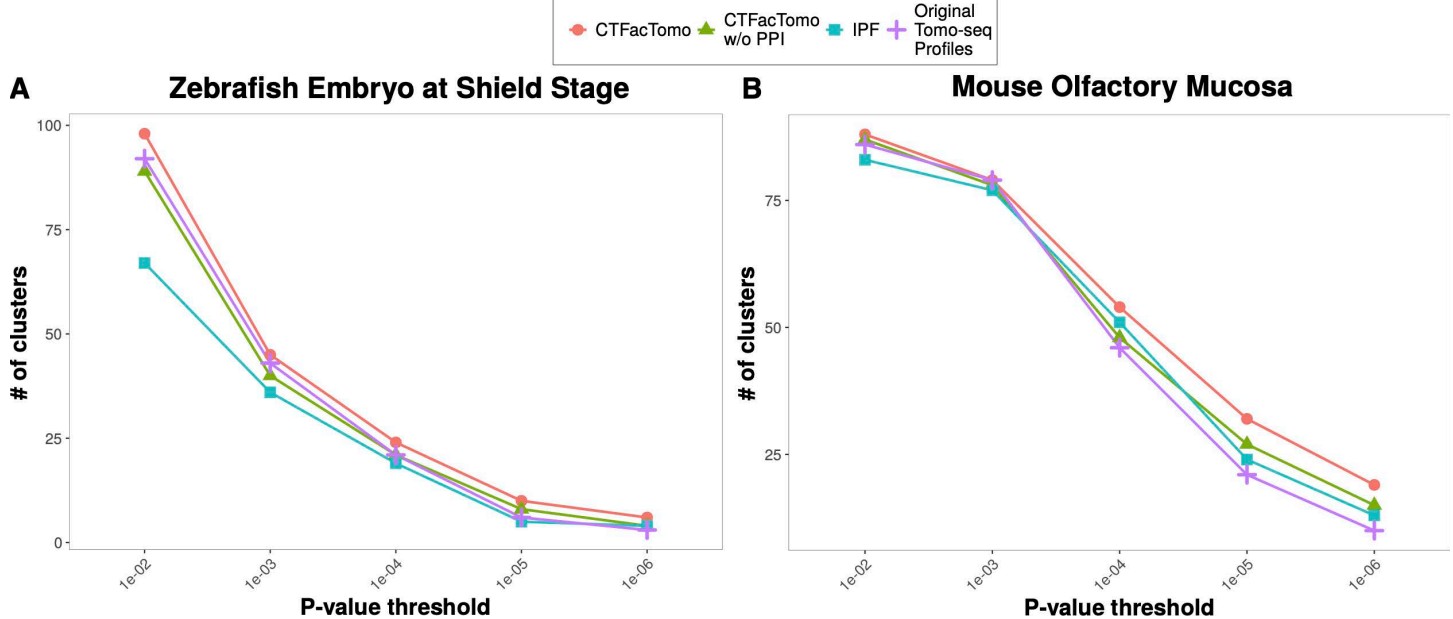

**Fig 8. Comparison of enriched gene clusters with reconstructed 3D expressions; Comparison of significantly enriched gene clusters identified using reconstructed 3D expressions.** The number of significantly enriched gene clusters at varying *p*-value thresholds in **A**: Zebrafish embryo and **B**: Mouse olfactory mucosa.

tomography profiles, which suggests CTFacTomo could improve the biological interpretation with more functional enrichment. Additionally, we introduced another comparison by replacing the PPI network in the product graph Laplacian regularization of CTFacTomo with an identity matrix of the same size. The results showed that CTFacTomo without the PPI network detected fewer significant gene clusters than CTFacTomo due to the loss of functional information in the PPI network. Nevertheless, this variation of CTFacTomo still detects overall more functionally enriched clusters than IPF in both datasets, and the original Tomo-seq profiles in mouse olfactory mucosa data. Note that this functional enrichment analysis is performed on gene clusters derived from whole transcriptome over the entire mixed cell populations, grouping genes with similar generic cellular functions. Varying the number of clusters primarily affects the granularity of Gene Ontology classifications, and the top enriched terms tend to correspond to generic cellular functions and biological processes, such as housekeeping and cellular cycle functions, rather than phenotype-specific or functionally distinctive biological signals.

## Discussion

Recently, microdissection-based methods such as variations of Tomo-seq, e.g., Geo-seq [40] and STRP-seq [16] have gained increasing attention. These newer technologies leverage adjacent slices as replicates such that no identical biological replicate is necessary for tomography. Furthermore, more flexible tomography views can be adopted beyond the three orthogonal views [16]. For instance, Geo-seq has been applied to explore temporal and spatial patterns in various embryo studies [41–44], and STRP-seq has demonstrated its potential in uncovering intricate spatial patterns in a non-model

organism [16]. These new developments greatly improved the applicability of microdissection-based methods. Additionally, high-throughput readout and relatively lower cost make these methods highly favorable for some specific research applications in developmental biology.

It is also important to note that CTFacTomo is based on a different formulation in contrast to the standard formulation of tensor factorization, where standard tensor factorization is derived from a given tensor with all (or some) known entries, while the formulation of CTFacTomo learns a factorization from the aggregated sums in certain views from collapsing the tensor without knowing any actual entry in the tensor to be constructed. Thus, the algorithm for such formulation is entirely different from standard tensor factorization methods and has a great potential to generalize to different kinds of tomography constructions in future work.

In general, reconstructing matrices or tensors from rank-1 summary statistics (e.g., spot-wise or gene-wise summation or averaging) is an ill-posed problem with many potentially divergent solutions. However, this challenge is substantially alleviated in spatial gene expression reconstruction, where expression patterns are constrained by underlying spatial structure rather than being arbitrarily independent across locations. Moreover, the 3D tensor formulation further restricts the solution space. For example, while matrix factorization is generally non-unique, the CP decomposition of a 3D tensor can be unique under certain rank conditions [23].

Importantly, CTFacTomo explicitly incorporates spatial information and gene functional relationships to effectively regularize the reconstruction. Spatial graph constraints favor spatially coherent expression patterns, while the PPI network aggregates information across functionally related genes, further restricting solutions through gene modules. As a result, CTFacTomo operates as a multitask framework that reconstructs the entire transcriptome jointly, in contrast to single-gene reconstruction methods such as IPF, which are more susceptible to noise. Consistently, our empirical results demonstrate that CTFacTomo outperforms IPF and Tomographer in 3D reconstruction tasks.

While the formulation of CTFacTomo requires a PPI network as input, the implementation allows replacement with a diagonal graph when such functional data is not available in some use cases. This simplification does not impose any functional information in the reconstruction and might reduce the functional interpretability of the reconstructed data but it will not greatly affect the reconstruction accuracy as we demonstrated in the experiments.

Finally, we note that CTFacTomo is not expected to perform optimally on 2D datasets, where constraints from only x–y projections are relatively weak. In contrast, Tomographer can leverage multiple projection views to achieve improved reconstruction of 2D patterns [16].

## Supporting information

**S1 Fig. Relationships between spot/gene density and spot/gene-wise $R^2$. A**: Projected human heart data **B**: Projected mouse brain data **C**: Projected Drosophila embryo data *left*: Scatter plot of spot-wise $R^2$ against spot density; *right*: Scatter plot of gene-wise $R^2$ against the rank of gene density. Each point represents an individual spot or gene, and the vertical lines in the gene-wise $R^2$ indicate gene density in the corresponding data. The dots that are at the minimum of $R^2$ are capped at the minimum for visualization, and thus are positioned along the horizontal line of the minimum $R^2$. (TIFF)

**S2 Fig. Comparison of reconstruction performance on the projected human heart data. A**: IPF (x-axis) and CTFacTomo (y-axis) **B**: Tomographer (x-axis) and CTFacTomo (y-axis) *left*: Scatter plot of spot-wise $R^2$ from different methods; *right*: Scatter plot of gene-wise $R^2$ from different methods. Each dot represents either a spot or a gene, and blue dots in both scatter plots indicate either the reconstructed expressions of spots or genes from CTFacTomo achieving higher $R^2$ compared to either IPF or Tomographer while red dots denote the reconstructed expressions of spots or genes from either IPF or Tomographer outperform CTFacTomo in terms of $R^2$. (TIFF)

**S3 Fig. Comparison of reconstruction performance on the projected mouse brain data.** A: IPF (x-axis) and CTFac-Tomo (y-axis) **B**: Tomographer (x-axis) and CTFacTomo (y-axis) *left*: Scatter plot of spot-wise $R^2$ from different methods; *right*: Scatter plot of gene-wise $R^2$ from different methods. Each dot represents either a spot or a gene, and blue dots in both scatter plots indicate either the reconstructed expressions of spots or genes from CTFacTomo achieving higher $R^2$ compared to either IPF or Tomographer while red dots denote the reconstructed expressions of spots or genes from either IPF or Tomographer outperform CTFacTomo in terms of $R^2$.
(TIFF)

**S4 Fig. Comparison of reconstruction performance on the projected Drosophila embryo data.** *left*: Scatter plot of spot-wise $R^2$ from Tomographer (x-axis) and CTFacTomo (y-axis); *right*: Scatter plot of gene-wise $R^2$ from Tomographer (x-axis) and CTFacTomo (y-axis). Each dot represents either a spot or a gene, and blue dots in both scatter plots indicate either the reconstructed expressions of spots or genes from CTFacTomo achieving higher $R^2$ compared to Tomographer while red dots denote the reconstructed expressions of spots or genes from IPF outperform either Tomographer in terms of $R^2$.
(TIFF)

**S5 Fig. Reconstruction loss of CTFacTomo under different ranks on two Tomo-seq datasets. A**: Zebrafish shield **B**: Mouse olfactory bulb. The vertical line indicates the rank determined by the heuristic method we implemented in the study.
(TIFF)

**S6 Fig. Moran scores between aligned slices for Stereo-seq validation.** Heatmap visualizations for the Moran scores between the reconstructed slices and Stereo-seq slices for five marker genes (*GSC*, *MAGI1B*, *MESPAB*, *NET1*, *INSB*). *left*: Scores when comparing CTFacTomo reconstructed slices with Stereo-seq slices. *right*: Scores when comparing IPF reconstructed slices with Stereo-seq slices. All spots within the reconstructed slice were considered. CTFacTomo always has a higher max score when comparing the heatmaps.
(TIFF)

**S7 Fig. Visualizations for the CTFacTomo reconstruction of the mouse olfactory mucosa on five marker genes.** The visualized slices are along the anteroposterior axis. The color scale ranges from blue to red, indicating low to high levels of relative expression for that gene, respectively. Gray indicates no expression.
(TIFF)

**S8 Fig. Visualizations for the IPF reconstruction of the mouse olfactory mucosa on five marker genes.** The visualized slices are along the anteroposterior axis. The color scale ranges from blue to red, indicating low to high levels of relative expression for that gene, respectively. Gray indicates no expression.
(TIFF)

**S9 Fig. Visual evaluations of the Moran's score on *ISG15* in Stereo-seq slice 2.** CTFacTomo scores higher by 0.105 when using all spots to calculate the Moran score. This is in part due to its capturing the most concentrated region of expression on the right side and the less pronounced region on the left. IPF misplaces its highest regions of expression in the wrong areas. In addition, CTFacTomo also weakly captured the center region.
(TIFF)

**S10 Fig. Visual evaluations of the Moran's score on *GSC*.** CTFacTomo scores higher by 0.171 when using all spots to calculate the Moran score. The highest area of expression in CTFacTomo aligns with the greatest area of expression in the Stereo-seq data. IPF is penalized by putting another region of high concentration on the wrong end of the reconstruction.
(TIFF)

**S11 Fig. Visual evaluations of the Moran's score on *ACTL6A*.** CFTactTomo scores higher by 0.13 when using all spots to calculate the Moran score. The spots are scattered across the whole tissue in the Stereo-seq data. CTFacTomo mostly captures both arms and the middle while IPF misses the middle.
(TIFF)

**S12 Fig. Visual evaluations of the Moran's score on *FOXA3*.** IPF scores higher by 0.20 when using all spots to calculate the Moran score. It is rewarded for putting some of its highest regions of expression in a similar area to the Stereo-seq. CTFacTomo is penalized for putting its highest region of expression at the center of the slice, where Stereo-seq shows no signal.
(TIFF)

**S13 Fig. Visual evaluations of the Moran's score on *ZGC:77262*.** CTFacTomo scores higher by 0.18 when using all spots to calculate the Moran score. Its regions of highest expression align with where Stereo-seq also has expression. IPF is penalized for putting its expression all over the place.
(TIFF)

**S14 Fig. Visual evaluations of the Moran's score on *RPA1*.** CTFacTomo scores higher by 0.25 when using all spots to calculate the Moran score. Its regions of highest expression align with where Stereo-seq also has expression. IPF is penalized for putting its expression all over the place.
(TIFF)

**S15 Fig. Visual evaluations of the Moran's score on *PMPCB*.** CTFacTomo scores higher by 0.183 when using all spots to calculate the Moran score. Its regions of highest expression align with where Stereo-seq also has expression. IPF is penalized for putting its expression all over the place.
(TIFF)

**S16 Fig. Visual evaluations of the Moran's score on *CCT6A*.** IPF scores higher here by 0.180 when using all spots to calculate the Moran score. The expression is scattered across the length of the Stereo-seq slice, a pattern which IPF captures whereas CTFacTomo incorrectly localizes its highest expression to the center.
(TIFF)

**S17 Fig. Visual evaluations of the Moran's score on *MBPB*.** IPF scores higher by 0.165 when using all spots to calculate the Moran score. Both IPF and CTFacTomo put too scattered expressions over the whole tissue while the expression in Stereo-seq focuses on the bottom arm. Since CTFacTomo had more wrong spots in the center and the top arm, it scored significantly worse. In addition, IPF also has better scatter in the bottom arm.
(TIFF)

**S18 Fig. Visual evaluations of the Moran's score on *STARD14*.** CTFacTomo scores higher by 0.23 when using all spots to calculate the Moran score. This can likely be attributed to CTFacTomo better capturing a high region of expression at the bottom arm of the slice. While IPF does also appear to capture some of the spatial patterns on the top and bottom arms, its highest region of expression is the incorrectly placed expression at the rightmost region of the slice.
(TIFF)

**S19 Fig. Visual evaluations of the Moran's score on *NOC3L*.** CTFacTomo scores higher by 0.20 when using all spots to calculate the Moran score. It's correct to put its highest region of expression to the right side of the slice, similar to Stereo-seq. IPF incorrectly places the highest points of expression in a chain along the center of the Stereo-seq slice.
(TIFF)

**S20 Fig. Visual evaluations of the Moran's score on *MID1IP1A*.** IPF scores higher here by 0.192 when using all spots to calculate the Moran score. IPF matches the Stereo-seq expression in that it spreads its expression all throughout the slice. While CTFacTomo shows expression in regions that do have expression, it is penalized for not capturing the entirety of the expression throughout the tissue.
(TIFF)

**S21 Fig. Visual evaluations of the Moran's score on *KRT4*.** IPF scores higher by 0.222 when using all spots to calculate the Moran score. Similar to the Stereo-seq, it represents its highest expression along the left and right edges with less expression at the center. CTFacTomo does capture some signal along the sides, but places its highest expression in the center and misses the transition regions between the center and the two arms.
(TIFF)

**S22 Fig. Visual evaluations of the Moran's score on *WNK4B* in Stereo-seq slice 9.** IPF scores higher by 0.22 when using all spots to calculate the Moran score. Likely, this results from IPF better capturing the expression regions at the top and bottom arms of the slice. Even though CTFacTomo does capture some expression in those regions as well, there are also more incorrectly placed regions of expression in CTFacTomo such as in the right-middle region of the slice, and also imprecise placement of expression in the top arm.
(TIFF)

**S23 Fig. Visual evaluations of the Moran's score on *ISG15* in Stereo-seq slice 9.** CTFacTomo scores higher by 0.37 when using all spots to calculate the Moran score. It accurately captures the concentrated regions of expression at the tips of the two arms as well as the center. IPF focuses its area of highest expression around the center, which is inconsistent with the Stereo-seq values.
(TIFF)

**S24 Fig. Visual evaluations of the Moran's score on *DPM3*.** CTFacTomo scores higher by 0.19 when using all spots to calculate the Moran score. It accurately captures a higher concentration of expression at the center of the slice, whereas IPF overspreads its expression.
(TIFF)

**S25 Fig. Visual evaluations of the Moran's score on *WNK4B* in Stereo-seq slice 12.** CTFacTomo scores higher by 0.363 when using all spots to calculate the Moran score. CTFaCtomo pinpoints the same region of high expression as Stereo-seq at the tip of top arm. IPF is penalized for putting its highest region of concentration on the lower right, where Stereo-seq has no expression at all.
(TIFF)

**S26 Fig. Visual evaluations of the Moran's score on *MAP12KB*.** CTFacTomo scores higher by 0.244 when using all spots to calculate the Moran score. It correctly places a concentrated region of expression on the top arm of the slice. The Moran's score penalizes IPF for overspreading its expression.
(TIFF)

**S27 Fig. ISH images for five marker genes in the mouse olfactory mucosa.** From top to bottom, the genes are *OLFR309*, *OLFR618*, *OLFR727*, *CYTL1*, *MOXD2*.
(TIFF)

## Acknowledgments

We sincerely thank Dr. Mayra Luisa, Dr. Antonio Scialdone, and Dr. Luis R. Saraiva for providing the mouse olfactory mucosa data and the scripts necessary to perform the experiments and the ISH images to evaluate the results.

## Author contributions

**Conceptualization:** Tianci Song, Rui Kuang.

**Data curation:** Tianci Song, Quoc Nguyen, Charles Broadbent.

**Formal analysis:** Tianci Song, Quoc Nguyen, Charles Broadbent, Rui Kuang.

**Funding acquisition:** Rui Kuang.

**Methodology:** Tianci Song, Quoc Nguyen, Rui Kuang.

**Software:** Tianci Song, Quoc Nguyen.

**Supervision:** Rui Kuang.

**Visualization:** Tianci Song, Quoc Nguyen.

**Writing – original draft:** Tianci Song, Quoc Nguyen, Rui Kuang.

**Writing – review & editing:** Tianci Song, Quoc Nguyen, Charles Broadbent, Rui Kuang.

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
