## [Decision Letter · Decision Letter 0]

25 Sep 2025

CTFacTomo: Reconstructing 3D Spatial Structures of RNA Tomography Transcriptomes by Collapsed Tensor Factorization

PLOS Computational Biology

Dear Dr. Kuang,

Thank you for submitting your manuscript to PLOS Computational Biology. After careful consideration, we feel that it has merit but does not fully meet PLOS Computational Biology's publication criteria as it currently stands.  In particular, both reviewers request substantial technical modifications to the manuscript to better substantiate your claims in terms of the algorithm's accuracy and performance, as well as clarify its limitations. Therefore, we invite you to submit a substantially revised version of the manuscript that addresses the points raised during the review process.

Please submit your revised manuscript within 60 days Nov 25 2025 11:59PM. If you will need more time than this to complete your revisions, please reply to this message or contact the journal office at ploscompbiol@plos.org. Please include the following items when submitting your revised manuscript:

We look forward to receiving your revised manuscript.

Kind regards,

Joshua N. Milstein

Academic Editor

PLOS Computational Biology

Shaun Mahony

Section Editor

PLOS Computational Biology

**Additional Editor Comments:**

Reviewer #1:

Reviewer #2:

**Journal Requirements:**

Potential Copyright Issues:

i) Figure 1. Please confirm whether you drew the images / clip-art within the figure panels by hand. If you did not draw the images, please provide (a) a link to the source of the images or icons and their license / terms of use; or (b) written permission from the copyright holder to publish the images or icons under our CC BY 4.0 license. Alternatively, you may replace the images with open source alternatives. See these open source resources you may use to replace images / clip-art:

2) If any authors received a salary from any of your funders, please state which authors and which funders..

7) Kindly revise your competing statement to align with the journal's style guidelines: 'The authors declare that there are no competing interests.'

**Reviewers' comments:**

Reviewer's Responses to Questions

**Comments to the Authors:**

Reviewer #1: The manuscript presents CTFacTomo, a collapsed-tensor factorization framework to reconstruct 3D spatial transcriptomes from RNA tomography (Tomo-seq) data using CP decomposition and a product graph Laplacian (gene PPI × 3 spatial chains). The method is evaluated on (i) simulated “projected” 3D ST datasets and (ii) two real Tomo-seq datasets (zebrafish embryo; mouse olfactory mucosa), with qualitative comparisons to ISH and a slice-level comparison to Stereo-seq using a bivariate Moran’s I.

1) Collapsing along two axes and learning CP factors is under-determined; please discuss conditions (rank, incoherence, graph priors, mask) under which the solution is identifiable or at least stable. Even a heuristic sensitivity study (vary rank, α, β; perturb inputs; re-seed) would help quantify solution variance.

2) Spatial graphs are described as chain graphs over slices. In real tissues, within-slice spatial adjacency (2D neighborhood) and anisotropic slice thickness matter. Please justify chain-only graphs vs. 3D voxel adjacency, and report results with a 3D 6-/18-/26-neighborhood graph to test robustness.

3) IPF and Tomographer are included, but other relevant baselines are missing: e.g., nonnegative tensor completion, graph-regularized matrix/tensor completion, SPICEMIX/GLUE-like graph models, or compressed sensing formulations. Even if adapted, adding one modern tensor completion baseline would strengthen claims.

4) More validations are encouraged, I suggest the authors add some more slide-level data used in this paper (https://doi.org/10.1038/s41467-025-63185-9) for comparisons.

4) Please provide open-source code, exact versions, and scripts to reproduce every figure/table, plus random seeds and data preprocessing (binning, normalization, PPI filtering). Without this, the work is difficult to verify.

Reviewer #2: Remarks to the Authors

This study ‘CTFacTomo: Reconstructing 3D Spatial Structures of RNA Tomography Transcriptomes by Collapsed Tensor Factorization’ by Song et al. introduces CTFacTomo which is a method for 3D reconstruction of gene expression from RNA tomography through regularized collapsed tensor factorization.

The technique’s core is a constrained Canonical Polyadic tensor decomposition framework in which latent spatial gene expression factors are inferred in such a way that their collapsed projections reproduce the observed slice-level transcirptomic profiles. The decomposition is regularized by biological priors which are derived from functional protein–protein interaction networks and by spatial prior through Laplacian smoothening.

The authors validate their method on spatial omics datasets of varying resolution and then benchmark it against two established RNA tomography pipelines, evaluating its performance based on similarity metrics of the reconstructed spatial signal and also based on the biological informativeness of the corresponding gene expression patterns.

The authors claim that CTFacTomo

1. Accurately reconstructs 3D gene expression from 1D bulk RNA-seq data

2. Is able to outperform Tomo-seq and Tomographer across datasets

3. Makes use of functional and spatial priors to resolve the underdetermined nature of the reconstruction

4. Generalizes algorithmically beyond transcriptomics, offering potential for other tomography tasks

I conclude that these claims are partially substantiated. Below, I provide some annotations for further analyses that are required to validate those claims (or alternatively how the limitations should be more clearly discussed).

In general, the technological approach is original, interesting and it is certainly a contribution to the current field of 3D spatial transcriptomics. The manuscript is clearly structured and well written. The figures are overall mostly accessible and informative to the reader. The manuscript’s method section is very well structured and the formulations are straight-forward and accessible, the git-hub repo provides an exhaustive tutorial and code annotations.

1. Method validation

For assessing their method’s reconstruction performance, the authors use data-set level MSE, MAE, spot-wise R2 and gene-wise R2 in their validation experiment. In the context of comparing 3D images this raises several questions in my opinion:

1.1 Please clarify why you chose MSE, MAE, spot-wise R2R^2R2, and gene-wise R2R^2R2 as the most appropriate choices for evaluating 3D spatial reconstructions.

1.2 Notably, in some cases the metrics seem to diverge. Intuitively, this does not necessarily mean the results are contradictory but it could also be the result of a complementary perspectives on reconstruction quality. I think a contextualization of what such divergences actually mean would be helpful to the reader. For example, could it be that high spot-wise R2R^2R2 but low gene-wise R2R^2R2 would suggest that the overall gene spectras at each spatial location are preserved but the spatial patterns of individual genes are blurred?

Additional to this contextualization, wouldn’t it be helpful to systematically addressed this, e.g. by providing analyses that explore source of such divergence (e.g. intensity scaling, spatial blurring, misalignment, statistical effects)?

1:3. Given the fact that the authors are working with synthetic tomography data from 3D reference, correlation-based measures (e.g. Pearson correlation per gene map or voxel spectrum) could be informative. This could help to capture agreement in spatial patterns ( how one gene varies across space) or spectral patterns (gene composition at one spatial location), independently from absolute intensity which could help the reader to better understand the provided error-based metrics (MSE/MAE) or some recommended additional image similarity metrics (SSIM or PSNR).

1.4. In the manuscript, the authors only provide dataset-level averages of benchmark metrics which does not provide details on variability. To complement this, I suggest to add distributions of metrics across the reconstructed genes. On this note, stratifying results by gene expression levels and/or variability/spatial complexity, would help us readers to assess where the method performs reliably and where its limitations lie more adequately.

1.5 For the reader to set the analyses described above in context, it would be helpful to have supplementary figure providing some visual reconstructions results against reference anecdotally.

1.6 The authors provide an ablation experiment showing that their priors improve their chosen performance metrics. It could be informative to include further controls (e.g. randomized PPI networks) to really demonstrate that gains are not due to generic regularization.

Furthermore, examining distributions and stratifications would again help to understand better whether improvements are really uniform or perhaps concentrated in some specific gene classes or spatial regions.

On that note, please also provide interpretation on how/why priors benefit the model (e.g. reducing outliers, improving genes of high variance, enhancing the joint spatial structuer of functionally related genes).

Comparison to Tomo-seq and Tomographer

In their second sets of experiment, the authors claim that CTFacTomo ‘performed better than IPF and Tomographer for 3D reconstruction’ and aim to support their claim by providing visual representations of the spatial reconstructions against a ground truth reference qualitatively and by quantifying spatial agreement through a bivariate Moran’s I comparison.

2.1 Visual inspection in Figure 4b shows that CTFacTomo is in fact able to recapitulate broad spatial patterns sucecssfully. However, given this visual inspection alone, I am not convinced that the statement can be made that CTFacTomo is generally more accurate than the Tomo-seq method. Instead, I believe there should be a more refined conclusion, ideally based on some additional systematic experiments.

In specific, thhere are some instances where the author’s method seems to capture signals that got lost by Tomo-seq (see gene GSC along the ventro-dorsal axis). However, in many other cases, the signal seems to be blurred in comparison to the Tomo-seq reconstruction result and also compred to the ISH data. Is there a way to systematically assess these positive and negative edge cases (or at least anecdotally assess by providing more and contextualized sets of visual inspection pairs)?

2.2 I agree with the authors in using bivariate Moran’s I as a means for quantitative comparison. The results look favorable for their method and provide support that CTFacTomo reconstructs signals successfully in spatial concordance with a reference.

However, taking into account that Moran’s I is consistently higher for their method than for IPF, while visual inspection does not really support this (see 2.1 above), I remain somewhat skeptical that the conclusion ‘higher Moran’s I = better reconstruction = superiority of CTFacTomo’ is justified. Specifically, given the blurring effects one can notice by visual inspecting the figures, I am afraid there is a real concern that the improved Moran’s I scores may be influenced at least in part by over-smoothing. The authors should therefore correct for this, e.g. provide complementary metrics or contextualize how oversmoothing might artificially inflate Moran’s I. On that note, please elaborate on the discrepancy between consistently high metrics scores and varying visual spatial concordance.

2.3 The claim that CTFacTomo performs better than Tomographer in 3D reconstructions is somewhat intuitive since Tomographer works in 2D. However, the conclusion that CTFacTomo is superior to two comparable methods in 3D (IPF and Tomographer) may be misleading without further context since Tomographer inherently does not seem to aim to preserve any 3D dependences and targets different use-cases than CTFacTomo or IPF. The authors should make this limitation even more explicit by discussing the comparison to IPF separately from the comparison to a 2D method.

2.4 Furthermore, the authors should provide visual inspection results for CTFacTomo against Tomographer.

2.5 In Figure 6 details of the ISH stainings are hardly readable. It would be helpful to have higher resolution images or rescaled versions.

2.6 From the comparisons mentioned above one can appreciate that there may be scenarios in which CTFacTomo may be a first-choice method, even if a generalized superiority of CTFacTomo should not be claimed. To highlight the benefits of you method, please discuss the use-cases in which CTFacTomo is the method of choice as opposed to scenarios in which other methods would likely be more adequate.

Functional and spatial priors

As mentioned in 1.6 the authors claim that spatial (Laplacian) and functional (PPI) priors for graph regularization improve their method’s performance. They then assess biological interpretability by clustering genes and then performing enrichment of GO terms, reporting the result that reconstructed 3D expressions yields more significant clusters than by IPF or raw 1D profiles.

3.1 Can you support your claim that more clusters = better biological interpretability (and not as a result of noise and overpartitioning which one could intuitively assume)?

3.2. On this note, please also provide some representative clusters with their spatial expression maps and enriched GO terms.

3.3. It should also be noted that the result of more significant GO clusters may be to some degree circular, since the model uses a PPI prior whereas GO enrichment tests may reflect similar functional relationships. Could you please discuss to which extend the reportet improvements in enrichment by PPI reg. actually reflects improved spatial reconstruction in contrast to some functional bias introduced by your chosen prior?

3.4 Replacing the PPI with an identity matrix demonstrates a scenario of having no functional prior. A pseudo PPI control could help to show that the positive effect is not due to generic regularization effects.

Algorithmic generalization

An exciting part of the manuscript lies in the fact that the authors developed an algorithm for tensor factorization that does depend on knowing any actual entries in the tensor to be constructed.

4.1 I believe this cornerstone method of your project deserves to be highlighted in more detail, e.g. providing a supplementary side-by-side method comparison and examining potential applications in the discussion part of your manuscript.

All in all, the authors provide a demonstration of their original 3D spatial signal reconstruction method in a well-written manuscript and contextualize its technological advantages by a set of biological experiments using simulated benchmarks and existing spatial transcriptomic data sets from different model animals.

I support a publication in principle pending revision as described above.

**Have the authors made all data and (if applicable) computational code underlying the findings in their manuscript fully available?**

Reviewer #1: None

Reviewer #2: Yes

PLOS authors have the option to publish the peer review history of their article (what does this mean? ). If published, this will include your full peer review and any attached files.

**Do you want your identity to be public for this peer review?** For information about this choice, including consent withdrawal, please see our Privacy Policy .

Reviewer #1: No

Reviewer #2: No

**Figure resubmission:**
---

## [Decision Letter · Decision Letter 1]

16 Feb 2026

Dear Prof. Kuang,

We are pleased to inform you that your manuscript 'CTFacTomo: Reconstructing 3D Spatial Structures of RNA Tomography Transcriptomes by Collapsed Tensor Factorization' has been provisionally accepted for publication in PLOS Computational Biology.

Best regards,

Joshua N. Milstein

Academic Editor

PLOS Computational Biology

Shaun Mahony

Section Editor

PLOS Computational Biology

Reviewer's Responses to Questions

**Comments to the Authors:**

Reviewer #1: I have no major concerns about this manuscript. But i didn't find the discussion regarding my previous critique on method evaluating on additional data and specific method (https://www.nature.com/articles/s41467-025-63185-9). I recommend the authors to address this concern too. Meanwhile, I suggest the authors to polish the whole manuscript and avoid any typos/grammar issues.

**Have the authors made all data and (if applicable) computational code underlying the findings in their manuscript fully available?**

Reviewer #1: None

PLOS authors have the option to publish the peer review history of their article (what does this mean? ). If published, this will include your full peer review and any attached files.

**Do you want your identity to be public for this peer review?** For information about this choice, including consent withdrawal, please see our Privacy Policy .

Reviewer #1: No

---

## [Editor Report · Acceptance letter]

PCOMPBIOL-D-25-01695R1

CTFacTomo: Reconstructing 3D Spatial Structures of RNA Tomography Transcriptomes by Collapsed Tensor Factorization

Dear Dr Kuang,

I am pleased to inform you that your manuscript has been formally accepted for publication in PLOS Computational Biology. Your manuscript is now with our production department and you will be notified of the publication date in due course.

With kind regards,

Anita Estes
